# ON ROLLOUTS IN MODEL-BASED REINFORCEMENT LEARNING

**Bernd Frauenknecht,**[*] **Devdutt Subhasish,**[*] **Friedrich Solowjow, and Sebastian Trimpe**
Institute for Data Science in Mechanical Engineering
RWTH Aachen University
Aachen, 52062, Germany
`firstname.lastname@dsme.rwth-aachen.de`

## ABSTRACT

Model-based reinforcement learning (MBRL) seeks to enhance data efficiency by learning a model of the environment and generating synthetic rollouts from it. However, accumulated model errors during these rollouts can distort the data distribution, negatively impacting policy learning and hindering long-term planning. Thus, the accumulation of model errors is a key bottleneck in current MBRL methods. We propose *Infoprop*, a model-based rollout mechanism that separates aleatoric from epistemic model uncertainty and reduces the influence of the latter on the data distribution. Further, Infoprop keeps track of accumulated model errors along a model rollout and provides termination criteria to limit data corruption. We demonstrate the capabilities of Infoprop in the *Infoprop-Dyna* algorithm, reporting state-of-the-art performance in Dyna-style MBRL on common MuJoCo benchmark tasks while substantially increasing rollout length and data quality.

## 1 INTRODUCTION

Reinforcement learning (RL) has emerged as a powerful framework for solving complex decision-making tasks like racing Vasco et al. (2024); Kaufmann et al. (2023) and gameplay OpenAI et al. (2019); Bi & D'Andrea (2024). However, when applying RL in real-world scenarios, a significant challenge is data inefficiency, which hinders the practicality of standard RL methods. Model-based reinforcement learning (MBRL) addresses this issue by learning an internal model of the environment Deisenroth & Rasmussen (2011); Chua et al. (2018); Janner et al. (2019); Hafner et al. (2020). By generating simulated interactions through model rollouts, MBRL can make informed decisions while substantially reducing the need for real-world data collection.

The quality of data from model-based rollouts is critical for MBRL performance. Model errors can distort the data distribution and hurt policy learning. Long-horizon planning is desirable, however, often infeasible as model errors accumulate over time. This effect is demonstrated in Figure 1. Even for a simple toy example (described in Appendix $B$), we see the data distribution of model-based rollouts under the state-of-the-art Trajectory Sampling (TS) Chua et al. (2018) scheme diverging quickly from the ground truth distribution of environment rollouts. Thus, data from TS rollouts can even be harmful to policy learning after a couple of time steps. This is largely because the TS mechanism does not explicitly address the effect of model errors on the propagated data distribution.

To tackle this challenge, we propose *Infoprop* , a novel model-based rollout mechanism that mitigates data distortion by addressing two key questions: *How to propagate?* and *When to stop?* We build our mechanism on explicitly leveraging the ability of common MBRL models to distinguish between aleatoric uncertainty due to process noise and epistemic uncertainty due to lack of data Lakshminarayanan et al. (2017); Becker & Neumann (2022). Making use of this property leads to substantially improved data consistency as depicted in Figure 1. In particular, we

- estimate and remove the stochasticity due to model error from the predictive distribution;
- formulate stopping criteria based on information loss to limit error accumulation; and

---

[*]Equal Contribution

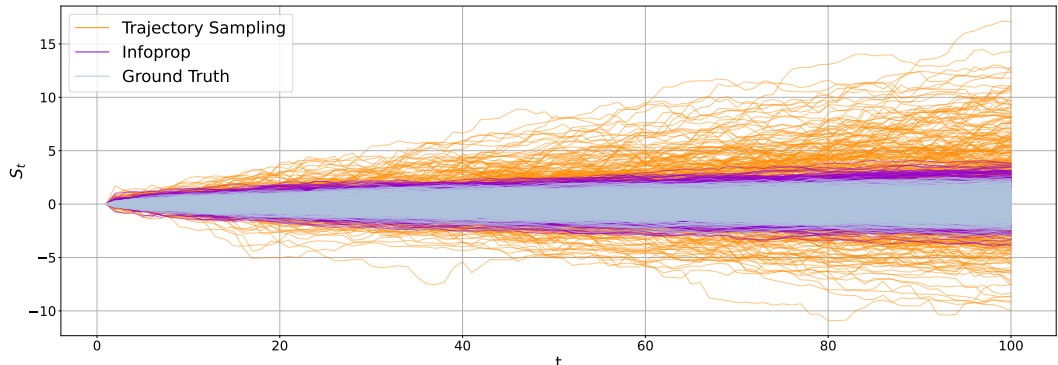

Figure 1: Comparing Data Consistency of Model-based Rollouts. *Trajectories under the proposed Infoprop mechanism follow the ground truth distribution of environment rollouts closely while rolling out the same model under the common TS scheme Chua et al. (2018) results in distorted data.*

- demonstrate the potential of Infoprop as a direct plugin to standard MBRL methods using the example of Dyna-style MBRL. The resulting *Infoprop-Dyna* algorithm yields state-of-the-art performance in MBRL on common MuJoCo tasks, while substantially improving the data consistency of model-based rollouts and thus allowing for longer rollout horizons.

## 2 BACKGROUND

In the following, we introduce the fundamental concepts of information theory and MBRL. Appendix A provides an overview of the notation introduced and used in the remainder of the paper.

### 2.1 INFORMATION THEORY

We will estimate the degree of data corruption in Infoprop rollouts using information-theoretic arguments. Information theory serves to quantify the uncertainty of a random variable (RV) Shannon (1948). Given the discrete RVs $X : \Omega \to \mathcal{X}$ and $Y : \Omega \to \mathcal{Y}$, the marginal entropy $\mathbb{H}(X) = -\sum_{x \in \mathcal{X}} \mathbb{P}[X = x] \log_2(\mathbb{P}[X = x])$ describes the average uncertainty about $X$ in bits. Further, the conditional entropy $\mathbb{H}(X|Y = y) = -\sum_{x \in \mathcal{X}} \mathbb{P}[X|Y = y] \log_2(P[X|Y = y])$ gives the uncertainty about $X$, given a realization of $Y$. Based on marginal and conditional entropy, the reduction in uncertainty about $X$ given a realization of $Y$ is described by mutual information

$$\mathbb{I}(X; Y = y) = \mathbb{H}(X) - \mathbb{H}(X|Y = y), \tag{1}$$

with $I(X; Y = y) = 0$ if the RVs are independent. In the following, we focus on Gaussian RVs and use the notion of quantized entropy Cover & Thomas (2006) with details provided in Appendix D.1.

### 2.2 REINFORCEMENT LEARNING

Reinforcement learning addresses sequential decision-making problems where the environment is typically modeled as a discrete-time Markov decision process (MDP) represented by the tuple $\mathcal{M} = \{\mathcal{S}, \mathcal{A}, \mathcal{R}, P_{\mathcal{R}}, P_{\mathcal{S}}, \xi_0, \gamma\}$. Here, $\mathcal{S} \subseteq \mathbb{R}^{n_{\mathcal{S}}}$ denotes the state space with $S_t \in \mathcal{S}$ being the RV of the state at time $t$ and $s_t$ its realization. Similarly, $\mathcal{A} \subseteq \mathbb{R}^{n_{\mathcal{A}}}$ represents the action space with $A_t \in \mathcal{A}$ the RV and $a_t$ the realization of the action as well as $\mathcal{R} \subseteq \mathbb{R}$ the set of rewards with $R_t \in \mathcal{R}$ and $r_t$ the reward at time $t$. We make the common simplifying assumption Bellemare et al. (2023) that the next state and reward are independent given the current state-action pair. Thus, a transition step in the environment can be expressed concerning a reward kernel $P_{\mathcal{R}}$ and a dynamics kernel $P_{\mathcal{S}}$ as

$$R_{t+1} \sim P_{\mathcal{R}}(\cdot|S_t, A_t) \quad \text{and} \quad S_{t+1} \sim P_{\mathcal{S}}(\cdot|S_t, A_t). \tag{2}$$

Further, initial states are distributed according to $S_0 \sim \xi_0$, and actions according to the policy $A_t \sim \pi(\cdot|S_t)$. We aim to learn an optimal policy $\pi^* = \arg\max_\pi \mathbb{E}_\pi \left[ \sum_{t=0}^\infty \gamma^t R_{t+1} \right]$ that maximizes the expected sum of rewards discounted by $\gamma \in [0, 1)$, referred to as return.

## 2.3 MODEL-BASED REINFORCEMENT LEARNING

There are four main categories of MBRL that all build on model-based rollouts. *(i)* Dyna-style methods Sutton (1991); Janner et al. (2019) use model-based rollouts to generate training data for a model-free agent. *(ii)* Model-based planning approaches Chua et al. (2018); Williams et al. (2017); Nagabandi et al. (2018); Hafner et al. (2019) do not learn an explicit policy but perform planning via model rollouts during deployment. *(iii)* Analytic gradient methods Deisenroth & Rasmussen (2011); Hafner et al. (2020; 2021; 2023) optimize the policy by backpropagating the performance gradient through model rollouts. *(iv)* Value-expansion approaches Feinberg et al. (2018); Buckman et al. (2018) stabilize the temporal difference target using model-based rollouts.

The model architecture of an MBRL algorithm determines the set of mechanisms for model rollouts. In this work, we focus on rolling out the particularly successful class of aleatoric epistemic separator (AES) models Lakshminarayanan et al. (2017); Becker & Neumann (2022) that can distinguish aleatoric uncertainty corresponding to the estimate of process noise from epistemic uncertainty.

## 2.4 ENVIRONMENT INTERACTION VS. MODEL-BASED ROLLOUTS

Model-based rollouts aim to substitute environment interaction in MBRL. Thus, we compare the data generation process through environment interaction to the process of model-based rollouts.

We model environment dynamics as a nonlinear function $\mu(S_t, A_t)$ with additive heteroscedastic process noise that is normally distributed with variance $\Sigma(S_t, A_t)$. Thus, environment rollouts, as depicted in Figure 1, are generated by iterating the dynamics

$$S_{t+1} = \mu(S_t, A_t) + L(S_t, A_t)W_t, \tag{3}$$

with $L(S_t, A_t)L(S_t, A_t)^\top = \Sigma(S_t, A_t)$ and the process noise $W_t \sim \mathcal{N}(0, I)$. Consequently, the transition kernel [1] of the environment is defined as $P_\mathcal{S}(\cdot|S_t, A_t) = \mathcal{N}(\mu(S_t, A_t), \Sigma(S_t, A_t))$.

In MBRL, however, we do not have access to $P_\mathcal{S}$ directly but typically rely on a parametric model with the random parameters $\Theta_t \in \vartheta$. Besides estimates of nonlinear dynamics $\hat{\mu}_{\Theta_t}(S_t, A_t)$ and process noise $\hat{\Sigma}_{\Theta_t}(S_t, A_t)$, AES models provide an estimate of the parameter distribution $\Theta_t \sim \mathbb{P}_\Theta$, e.g. via ensembling Lakshminarayanan et al. (2017) or dropout Becker & Neumann (2022). These models are typically propagated using the TS Chua et al. (2018) rollout mechanism via iterating

$$S_{t+1} = \hat{\mu}_{\Theta_t}(S_t, A_t) + \hat{L}_{\Theta_t}(S_t, A_t)W_t \tag{4}$$

with $\hat{L}_{\Theta_t}(S_t, A_t)\hat{L}_{\Theta_t}(S_t, A_t)^\top = \hat{\Sigma}_{\Theta_t}(S_t, A_t)$, $W_t \sim \mathcal{N}(0, I)$, and $\Theta_t \sim \mathbb{P}_\Theta$. This results in the TS rollouts in Figure 1 and induces the kernel $\hat{P}_{\mathcal{S},\mathrm{TS}}(\cdot|S_t, A_t) = \mathcal{N}\left(\hat{\mu}_{\Theta_t}(S_t, A_t), \hat{\Sigma}_{\Theta_t}(S_t, A_t)\right)$. The majority of recent MBRL approaches use the TS rollout mechanism, e.g. Chua et al. (2018); Becker & Neumann (2022); Janner et al. (2019); Pan et al. (2020); Yu et al. (2020); Luis et al. (2023). Pseudocode is provided in Algorithm 2 of Appendix C.

## 3 PROBLEM STATEMENT

Revisiting Figure 1 allows us to illustrate the effects of different sources of stochasticity by comparing environment interaction under $P_\mathcal{S}$ to TS rollouts under $\hat{P}_{\mathcal{S},\mathrm{TS}}$. While different realizations of process noise $w_t \sim \mathcal{N}(0, I)$ allow for keeping track of the environment distribution, the sampling process $\theta_t \sim \mathbb{P}_\Theta$ introduces additional stochasticity that leads to an overestimated total variance in the TS rollout distribution. This effect is amplified through the continued propagation of erroneous predictions making data at later steps unfit for policy learning. We ask the following questions:

(i) How can we construct a predictive distribution closely resembling environment dynamics?
(ii) How can we quantify the degree of data corruption due to model error?
(iii) When should model-based rollouts be terminated due to data corruption?

We address these questions by proposing the *Infoprop* rollout mechanism. Infoprop isolates and removes epistemic uncertainty for an improved predictive distribution, keeps track of data corruption using information-theoretic arguments, and terminates rollouts based on the degree of corruption.

---

[1] As $P_\mathcal{R}$ typically is a known deterministic function in the context of MBRL, while $P_\mathcal{S}$ is the unknown object we aim to model, the discussion henceforth focuses on approximating $P_\mathcal{S}$ without loss of generality.

## 4 INFOPROP ROLLOUT MECHANISM

In the following, we introduce the Info-prop mechanism for model-based rollouts. As depicted in Figure 2, we decompose model predictions into a signal fraction representing the environment dynamics and noise fraction introduced by model error. This perspective allows to interpret model rollouts as communication through a noisy channel. We estimate both the signal and the noise distribution and use these to infer a belief over the environment state, given an observation of the model state. This belief state represents the foundation of the Infoprop rollout mechanism.

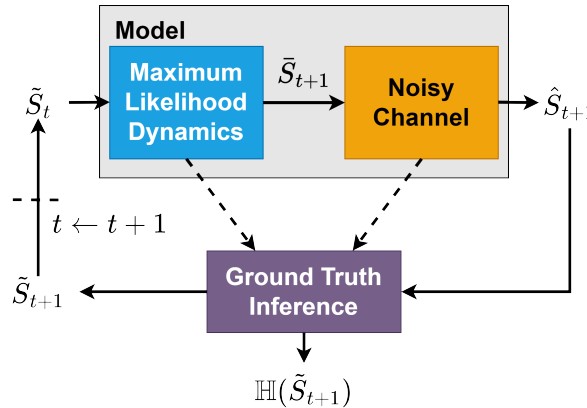

Figure 2: Infoprop block diagram

### 4.1 THEORETICAL SETUP

First, we introduce additional notation to specify RVs under different transition kernels.

**Definition 1** (Environment state). *We define the environment state as the conditional expectation under the environment dynamics given a realization of a state-action pair*

$$\check{S}_{t+1} := \mathbb{E}_{P_{\mathcal{S}}}\left[S_{t+1}|S_t = s_t, A_t = a_t, W_t\right]. \tag{5}$$

Thus, $\check{S}_{t+1}$ is an RV, where the randomness is induced by the process noise and has an aleatoric nature. If we additionally condition on the realization $W_t = w_t$, we obtain a deterministic object.

**Definition 2** (Model state). *We define the model state as the conditional expectation under $\hat{P}_{\mathcal{S},\mathrm{TS}}$*

$$\hat{S}_{t+1} := \mathbb{E}_{\hat{P}_{\mathcal{S},\mathrm{TS}}}\left[S_{t+1}|S_t = s_t, A_t = a_t, W_t, \Theta_t\right]. \tag{6}$$

As discussed in Section 3, stochasticity in $\hat{S}_{t+1}$ is induced not only by $W_t$ but also by the randomness in the parameters $\Theta_t$. We project the uncertainty in the parameter space $\vartheta$ to $\mathcal{S}$ via an error process.

**Definition 3** (Model error process). *We define a model error process*

$$\Delta_t := \hat{S}_{t+1} - \check{S}_{t+1} \tag{7}$$

*that, given a realization of process noise $W_t = w_t$, projects uncertainty in $\vartheta$ to $\mathcal{S}$*

$$\mathbb{E}\left[\Delta_t|W_t = w_t\right] = \mathbb{E}_{\hat{P}_{\mathcal{S},\mathrm{TS}}}\left[S_{t+1}|s_t, a_t, w_t, \Theta_t\right] - \mathbb{E}_{P_{\mathcal{S}}}\left[S_{t+1}|s_t, a_t, w_t\right]. \tag{8}$$

*We refer to the projected parameter uncertainty as epistemic uncertainty.*

Further, we restrict model usage to a sufficiently accurate subset $\mathcal{E} \subseteq \mathcal{S} \times \mathcal{A}$, as proposed in Frauenknecht et al. (2024). We define $\mathcal{E}$ amenable to the Infoprop setting in Section 4.4 and make the following assumptions when performing model-based rollouts in $\mathcal{E}$:

**Assumption 1** (Consistent estimator of aleatoric uncertainty). *The model's predictive variance $\hat{\Sigma}_{\Theta_t}$ is a consistent estimator of $\Sigma$ following the definition of Julier & Uhlmann (2001), i.e.*

$$\left(\hat{\Sigma}_{\Theta_t}(S_t, A_t) - \Sigma(S_t, A_t)\right) \succcurlyeq 0 \quad \forall(S_t, A_t) \in \mathcal{E}. \tag{9}$$

**Assumption 2** (Unbiased estimator). *The model bias $\mu^\Delta$ is negligible. Thus $\hat{S}_{t+1}$ according to (4) is an unbiased estimator of $\check{S}_{t+1}$ according to (3), i.e.*

$$\mathbb{E}[\hat{S}_{t+1}|S_t, A_t] = \mathbb{E}\left[\check{S}_{t+1}|S_t, A_t\right] \quad \forall(S_t, A_t) \in \mathcal{E}. \tag{10}$$

Figure 1 empirically shows that these assumptions are reasonable. The Infoprop distribution is slightly more stochastic than the ground truth process, which indicates that Assumption 1 holds. As (9) states, the model does not underestimate aleatoric uncertainty; the Infoprop rollouts should be at least as stochastic as the true process. Further, we observe no substantial bias of the Infoprop distribution underscoring the soundness of Assumption 2. Infoprop shows a similar behavior in high dimensional problems as reported in Section 6.

## 4.2 DECOMPOSING THE MODEL STATE IN SIGNAL AND NOISE

We aim to isolate the stochasticity due to parameter uncertainty in $\hat{S}_{t+1}$. We use the model error process (8) to project the noise in $\vartheta$ to the same space as the signal, i.e. the dynamics, which is $\mathcal{S}$. The parameter distribution $\Theta_t \sim \mathbb{P}_\Theta$ can induce arbitrarily complex distributions $\Delta_t \sim \mathbb{P}_\Delta$. To simplify the analysis, we solely consider the first two moments of $\mathbb{P}_\Delta$, namely $\mu^\Delta$ and $\Sigma^\Delta$. This allows to reformulate the propagation equation (4) of the model state

$$\hat{S}_{t+1} = \check{S}_{t+1} + \Delta_t \approx \check{S}_{t+1} + \mu^\Delta(S_t, A_t) + L^\Delta(S_t, A_t)N_t \qquad (11)$$

concerning $\check{S}_{t+1}$ and the model error $\Delta_t$ represented by $\mu^\Delta(S_t, A_t)$ the model bias, $\Sigma^\Delta(S_t, A_t)$ the epistemic variance with Cholesky decomposition $L^\Delta(S_t, A_t)$, and $N_t$ the epistemic noise.

By Assumption 2, we have $\mu^\Delta(S_t, A_t) = 0 \quad \forall (S_t, A_t) \in \mathcal{E}$. Consequently, we can interpret the model rollout as communication through a Gaussian noise channel Cover & Thomas (2006) via (11).

Based on the propagation equation (11), we aim to infer the maximum likelihood estimate of $\check{S}_{t+1}$ from $E$ realizations of $\{\mathbb{E}[\hat{S}_{t+1}|N_t = n_t^e]\}_{e=1}^E$, to use it as the predictive distribution for our rollout scheme. As we cannot sample $N_t$ directly, we instead use an equivalent definition of $\hat{S}_{t+1}$.

**Definition 4** (Model state concerning epistemic uncertainty). *Based on the model error process (8) the model state is defined as*

$$\hat{S}_{t+1} = \mathbb{E}_{\hat{P}_{\mathcal{S},\mathrm{TS}}}[S_{t+1}|S_t = s_t, A_t = a_t, W_t, \Delta_t] \approx \mathbb{E}_{\hat{P}_{\mathcal{S},\mathrm{TS}}}[S_{t+1}|S_t = s_t, A_t = a_t, W_t, N_t] \quad (12)$$

Reformulating (6) concerning $\Delta_t$ does not change the information content or the induced sigma-algebra, as $\Delta_t$ is a measurable function of $\Theta_t$. In the simplified setting of solely considering the first two moments of $\mathbb{P}_\Delta$, $N_t$ fully describes stochasticity due to model error. In reverse, we can obtain realizations $\{\mathbb{E}[\hat{S}_{t+1}|\Theta_t = \theta_t^e]\}_{e=1}^E$ and interpret them as samples $\{\mathbb{E}[\hat{S}_{t+1}|N_t = n_t^e]\}_{e=1}^E$.

**Lemma 1.** *Given $E$ realizations of $\mathbb{E}\left[\hat{S}_{t+1}|\Theta_t = \theta_t^e\right]$, we can estimate the environment state using maximum likelihood as*

$$\check{S}_{t+1} = \mathbb{E}\left[\hat{S}_{t+1}|N_t = 0\right] \approx \bar{S}_{t+1} = \bar{\mu}(S_t, A_t) + \bar{L}(S_t, A_t)W_t \qquad (13)$$

*Proof.* see Appendix D.2.1 □

**Lemma 2.** *Following this line of thought, the maximum likelihood estimate of $\Sigma^\Delta$ is given by*

$$\bar{\Sigma}^\Delta(S_t, A_t) = \frac{1}{E} \sum_{e=1}^E \left(\hat{\mu}_{\Theta_t=\theta_t^e}(S_t, A_t) - \bar{\mu}(S_t, A_t)\right) \left(\hat{\mu}_{\Theta_t=\theta_t^e}(S_t, A_t) - \bar{\mu}(S_t, A_t)\right)^\top. \quad (14)$$

*Proof.* see Appendix D.2.2 □

Given the maximum likelihood estimates of the environment state $\bar{S}_{t+1}$ and the epistemic variance $\bar{\Sigma}^\Delta$, we can decompose the model state $\hat{S}_{t+1}$ in a signal and noise fraction according to (11) in $\mathcal{E}$.

## 4.3 CONSTRUCTING THE INFOPROP STATE

Having decomposed $\hat{S}_{t+1}$ into signal $\bar{S}_{t+1}$ and noise $\bar{\Sigma}^\Delta$, allows us to define the Infoprop state.

**Definition 5** (Infoprop state). *We define the Infoprop state*

$$\tilde{S}_{t+1} := \mathbb{E}\left[\bar{S}_{t+1}|\hat{S}_{t+1} = \hat{s}_{t+1}\right] = \mathbb{E}_{\tilde{P}_{\mathcal{S},\mathrm{IP}}}\left[S_{t+1}|S_t = s_t, A_t = a_t, \hat{S}_{t+1} = \hat{s}_{t+1}, U_t\right] \quad (15)$$

*as the conditional expectation of the estimated environment state given a sample of the model state. We derive the corresponding Infoprop kernel $\tilde{P}_{\mathcal{S},\mathrm{IP}}(\cdot|S_t, A_t, \hat{S}_{t+1}) = \mathcal{N}\left(\tilde{\mu}(S_t, A_t, \hat{S}_{t+1}), \tilde{\Sigma}(S_t, A_t, \hat{S}_{t+1})\right)$ with the conditional noise $U_t \sim \mathcal{N}(0, I)$ in Appendix D.3.*

Consequently, the Infoprop state aims to infer the signal $\bar{S}_{t+1}$ given a noisy observation $\hat{s}_{t+1}$. Propagating model-based rollouts using $\tilde{S}_{t+1}$, yields favorable properties as stated in Theorem 1.

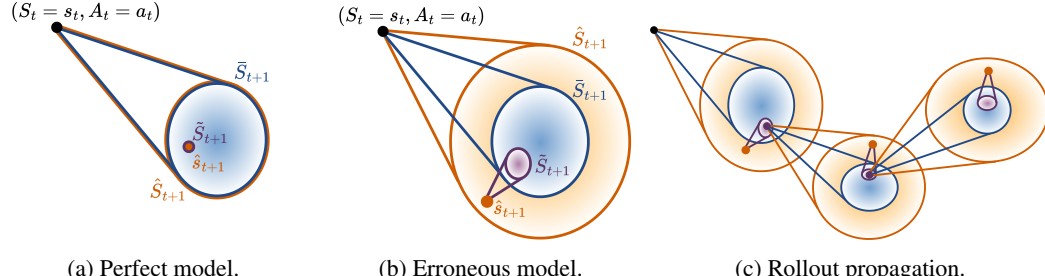

|  |  |  |
|---|---|---|
| (a) Perfect model. | (b) Erroneous model. | (c) Rollout propagation. |

Figure 3: Infoprop rollout mechanism. *(a), (b): Generating the Infoprop state $\tilde{S}_{t+1}$ from the estimated predictive distribution $\bar{S}_{t+1}$ and the model sample $\hat{s}_{t+1}$. (c) Performing an Infoprop rollout.*

**Theorem 1** (Infoprop state). *By construction, $\tilde{S}_{t+1}$ addresses questions (i) and (ii) of Section 3.*

  (i) *The distribution of Infoprop states is identical to the estimated environment distribution.*

$$\tilde{S}_{t+1} \stackrel{\text{dist}}{=} \bar{S}_{t+1} \tag{16}$$

  *Proof.* see Appendix D.4. □

  (ii) *The sum of marginal entropies of $\tilde{S}_{t+1}$ defines the information loss along an Infoprop rollout.*

$$\mathbb{H}\left(\bar{S}_1, \bar{S}_2, \ldots \bar{S}_T | S_0 = s_0, A_0 = a_0, \hat{S}_1 = \hat{s}_1, \ldots \hat{S}_T = \hat{s}_T\right) = \sum_{t=0}^{T} \mathbb{H}\left(\tilde{S}_{t+1}\right) \tag{17}$$

  *Proof.* see Appendix D.5. □

Figure 3 illustrates the Infoprop rollout mechanism and provides intuition for Theorem 1. In the case of a perfect model, i.e. $\bar{\Sigma}^\Delta = 0$, depicted in Figure 3a, the realization $\hat{s}_{t+1}$ provides the information about the process noise realization $w_t$ without ambiguity. Consequently, the belief about the environment state given the sample from the model $\tilde{S}_{t+1} = \mathbb{E}[\bar{S}_{t+1} | \hat{S}_{t+1} = \hat{s}_{t+1}]$ is a deterministic object and $\mathbb{H}(\tilde{S}_{t+1}) = 0$. In the general scenario of $\bar{\Sigma}^\Delta > 0$ depicted in Figure 3b, the epistemic uncertainty results in ambiguity about the environment state given $\hat{s}_{t+1}$, such that $\mathbb{H}(\tilde{S}_{t+1}) > 0$. Notably, conditioning $\bar{S}_{t+1}$ on $\hat{s}_{t+1}$, results in Infoprop predictions $\tilde{S}_{t+1}$ following estimated environment distribution $\bar{S}_{t+1}$ as stated in Theorem 1 (i). This results in a data distribution that closely resembles the environment dynamics as desired in question (i) of Section 3. Finally, Figure 3c depicts a Infoprop rollout propagated via realization $\tilde{s}_{t+1}$. We measure data corruption due to model error using the conditional entropy of a rollout under the estimated environment dynamics $(\bar{S}_1, \bar{S}_2, \ldots \bar{S}_T)$ given the realizations observed from the model $(S_0 = s_0, A_0 = a_0, \hat{S}_1 = \hat{s}_1, \ldots \hat{S}_T = \hat{s}_T)$, i.e. *given the observed model trajectory, how sure are we on how the corresponding environment trajectory would look like?*. As per Theorem 1 (ii), this trajectory-based approach to uncertainty can be addressed with the accumulated marginal entropy of $\tilde{S}_{t+1}$, addressing question (ii) of Section 3.

## 4.4 ROLLOUT TERMINATION CRITERIA

Having introduced how to propagate Infoprop rollouts, the question remains when to terminate them. In the following, we propose two termination criteria to address question (iii) of Section 3.

First, Infoprop rollouts build on the assumption that model usage is restricted to a sufficiently accurate subset $\mathcal{E} \subseteq \mathcal{S} \times \mathcal{A}$, following the ideas of Frauenknecht et al. (2024).

**Definition 6** (Sufficiently accurate subset). *We define the sufficiently accurate subset*

$$\mathcal{E} := \{(s_t, a_t) \in \mathcal{S} \times \mathcal{A} \mid \mathbb{H}(\tilde{S}_{t+1}) \le \lambda_1, \hat{s}_{t+1} \sim \hat{P}_{\mathcal{S},\text{TS}}(\cdot | s_t, a_t)\} \tag{18}$$

*based on a threshold $\lambda_1$ for the single-step information loss $\mathbb{H}(\tilde{S}_{t+1})$.*
Second, we restrict Infoprop rollouts to sufficiently accurate paths to limit uncertainty accumulation.

**Definition 7** (Sufficiently accurate path). *Based on the estimated information loss along a rollout (17), we define the set of sufficiently accurate paths of length $t' \in \{1, \dots, T\}$ as*

$$\mathcal{P}^{t'} := \left\{ (s_t, a_t)_{t=0}^{t'} \in (\mathcal{S} \times \mathcal{A})^{t'} \,\middle|\, \sum_{t=0}^{t'} \mathbb{H}(\tilde{S}_{t+1}) \leq \lambda_2 \right\}. \tag{19}$$

Heuristics for determining values of $\lambda_1$ and $\lambda_2$ depend on the class of AES model and MBRL algorithm at hand with an example provided in Section 5. Combining the steps above yields the Infoprop rollout mechanism illustrated in Algorithm 1.

---

**Algorithm 1** Infoprop

---

**Require:** $s_0$
  **while** $t < T + 1$ **do**
    $a_t \sim \pi(\cdot|s_t)$
    **for** $e \in \{1, \dots, E\}$ **do**
      $\theta_t^e \sim \mathbb{P}_\Theta$
    $\bar{S}_{t+1}(s_t, a_t)$ from (13), and $\bar{\Sigma}^\Delta(s_t, a_t)$ from (14)
    $\hat{s}_{t+1} = \mathbb{E}\left[\hat{S}_{t+1}|W_t = w_t, \Theta_t = \theta_t^{e'}\right]$ with $w_t \sim \mathcal{N}(0, I), \theta_t^{e'} \sim \mathcal{U}(\{\theta_t^1, \dots, \theta_t^E\})$
    $\tilde{S}_{t+1}$ from (51) and $\mathbb{H}(\tilde{S}_{t+1})$ from (24)
    **if** $\mathbb{H}(\tilde{S}_{t+1}) > \lambda_1$ **then**
      **break**
    **else if** $\sum_{t'=0}^{t} \mathbb{H}(\tilde{S}_{t'+1}) > \lambda_2$ **then**
      **break**
    **else**
      $s_t \leftarrow \mathbb{E}[\tilde{S}_{t+1}|U_t = u_t]$ with $u_t \sim \mathcal{N}(0, I)$

---

## 5   Augmenting State-Of-The-Art: Infoprop-Dyna

While the Infoprop rollout mechanism is applicable to different kinds of MBRL with AES models, we illustrate its capabilities in a Dyna-style architecture with probabilistic ensemble (PE) models Lakshminarayanan et al. (2017). We design *Infoprop-Dyna* by integrating the Infoprop rollout mechanism in the state-of-the-art framework proposed in Janner et al. (2019) with minor adaptions.

As discussed in Section 4.4, heuristics for $\lambda_1$ and $\lambda_2$ depend on the algorithm at hand. In Infoprop-Dyna, we take the common approach Chua et al. (2018); Janner et al. (2019) of neglecting cross-correlations between state dimensions for computational reasons. Thus, we can consider data corruption of each state dimension independently. As the predictive quality of different state dimensions can differ substantially, we choose both thresholds as $n_\mathcal{S}$ dimensional vectors, such that a rollout is terminated as soon as the data corruption of any dimension overshoots the corresponding threshold.

In Dyna-style MBRL Janner et al. (2019), the dynamics model is trained on the data distribution observed during environment interaction. The corresponding transitions are stored in an environment replay buffer $\mathcal{D}_{\text{env}} = \{(\check{s}_t^{(b)}, \check{a}_t^{(b)}, \check{r}_{t+1}^{(b)}, \check{s}_{t+1}^{(b)})\}_{b=1}^{|\mathcal{D}_{\text{env}}|}$, where $(b)$ indicates the index in the replay buffer. After a fixed number of interaction steps between a model-free RL agent and the environment, the dynamics model is retrained on the data in $\mathcal{D}_{\text{env}}$, model-based rollouts are performed, and the data is stored stored in a replay buffer $\mathcal{D}_{\text{mod}}$ to train the model-free RL agent. Consequently, we assume the PE model to be accurate within the data distribution of $\mathcal{D}_{\text{env}}$ and build the heuristic for $\lambda_1$ and $\lambda_2$ on the predictive uncertainty within the environment buffer.

After each round of retraining the PE model, we compute a set of dimension-wise Infoprop state entropies for single-step predictions in $\mathcal{D}_{\text{env}}$ according to

$$\mathcal{H}^k = \left\{ \mathbb{H}\left( \bar{S}_{t+1}^k | S_t = \check{s}_t^{(b)}, A_t = \check{a}_t^{(b)}, \hat{S}_{t+1}^k = \hat{s}_{t+1}^{k,(b)} \right) = \mathbb{H}\left( \tilde{S}_{t+1}^{k,(b)} \right) \right\}_{b=1}^{|\mathcal{D}_{\text{env}}|} \tag{20}$$

where $k \in \{1, \ldots, n_{\mathcal{S}}\}$ indicates the corresponding state dimension. We define the dimension-wise thresholds $\lambda_1^k$ and $\lambda_2^k$ based on the cumulative distribution function of dimension-wise entropies

$$F_{\mathcal{H}^k}(h) = \frac{1}{|\mathcal{H}^k|} \sum_{h' \in \mathcal{H}^k} \mathbb{1}[h' \leq h]. \tag{21}$$

The $k^{\text{th}}$ element of $\lambda_1$ is defined as the $\zeta_1$ quantile of the single-step entropy set

$$\lambda_1^k = \inf \left\{ h \in \mathcal{H}^k : F_{\mathcal{H}^k}(h) \geq \zeta_1 \right\} \tag{22}$$

and limits model usage to the sufficiently accurate subset $\mathcal{E}$. To restrict rollouts of length $t'$ to $\mathcal{P}^{t'}$, we define the $k^{\text{th}}$ element of $\lambda_2$ as the $\zeta_2$ quantile of the entropy set scaled by $\xi$

$$\lambda_2^k = \xi \inf \left\{ h \in \mathcal{H}^k : F_{\mathcal{H}^k}(h) \geq \zeta_2 \right\}. \tag{23}$$

Here, $\zeta_2$ denotes a quantile corresponding to precise predictions and $\xi$ to the number of prediction steps we are willing to accumulate the resulting data corruption. We choose $\zeta_1 = 0.99$, $\zeta_2 = 0.01$ and $\xi = 100$ for all experiments in Section 6 without further hyperparameter tuning.

We use pink noise for environment exploration Eberhard et al. (2023) to quickly expand $\mathcal{E}$ Frauenknecht et al. (2024). Pseudocode is provided in Algorithm 3 of Appendix C.

## 6 EXPERIMENTS AND DISCUSSION

To demonstrate the benefits of the Infoprop mechanism, we compare Infoprop-Dyna to state-of-the-art Dyna-style MBRL algorithms on MuJoCo Todorov et al. (2012) benchmark tasks. We report

- substantial improvements in the consistency of predicted data, especially over long horizons;
- effective rollout termination based on accumulated model error propagation; and
- state-of-the-art performance in Dyna-style MBRL on several MuJoCo tasks.

Furthermore, we discuss the limitations of naively integrating Infoprop into the standard Dyna-style setup Janner et al. (2019) and point to further research questions.

### 6.1 EXPERIMENTAL SETUP

We compare Infoprop-Dyna to Model-Based Policy Optimization (MBPO) Janner et al. (2019) and Model-Based Actor-Critic with Uncertainty-Aware Rollout Adaption (MACURA) Frauenknecht et al. (2024) as well as to Soft Actor-Critic (SAC) Haarnoja et al. (2018) that represents the model-free learner of all the Dyna-style approaches above. We build our implementation[2] on the code base[3] provided by Frauenknecht et al. (2024). Further details are provided in Appendix E.1

### 6.2 PREDICTION QUALITY

To compare different rollout mechanisms, we train an Infoprop-Dyna agent on hopper for 120000 environment interactions and perform model rollouts from states in $\mathcal{D}_{\text{env}}$.

First, we evaluate the consistency of Infoprop and TS rollouts, propagating 20 steps without termination. Figure 4a depicts the resulting distributions for the $11^{\text{th}}$ dimension of the hopper state. Infoprop rollouts show substantially improved data consistency compared to TS rollouts, underscoring the ability of Infoprop to effectively mitigate model error propagation.

Next, we compare the rollout mechanisms of MBPO and MACURA based on TS sampling with Infoprop-Dyna rollouts. Figure 4b shows the results for $11^{\text{th}}$ dimension of the hopper and a maximum rollout length of 100 steps. MBPO rollouts are propagated for 11 steps following the schedule proposed in Janner et al. (2019), resulting in a widely spread distribution. In contrast, MACURA has an adaptive rollout length capped at 10 steps Frauenknecht et al. (2024), leading to better data consistency. The improved predictive distribution and capability to estimate accumulated error of Infoprop allows for substantially longer rollouts up to 100 steps. The Infoprop termination criteria reliably stop distorted rollouts, resulting in consistent rollouts over long horizons. Appendix E.2 provides additional results for setting the maximum rollout length of all three approaches to 100.

---

[2]https://github.com/Data-Science-in-Mechanical-Engineering/infoprop
[3]https://github.com/Data-Science-in-Mechanical-Engineering/macura

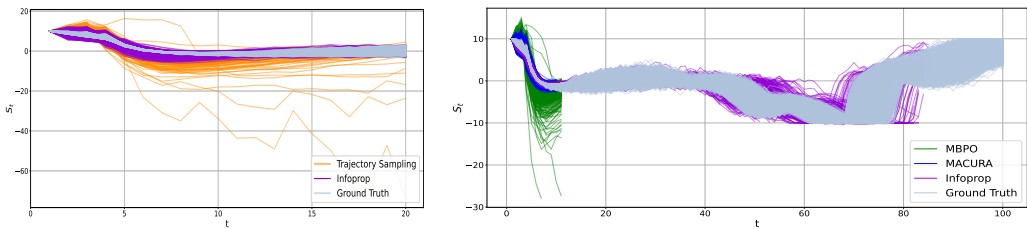

(a) Trajectory Sampling vs. Infoprop      (b) MBPO vs. MACURA vs. Infoprop-Dyna

Figure 4: Predictive quality of rollouts in the $11^{th}$ state dimension of MuJoCo hopper. *(a) Rollouts according to Trajectory Sampling (TS) and Infoprop . (b) Rollout schemes of MBPO and MACURA based on TS compared to Infoprop-Dyna .*

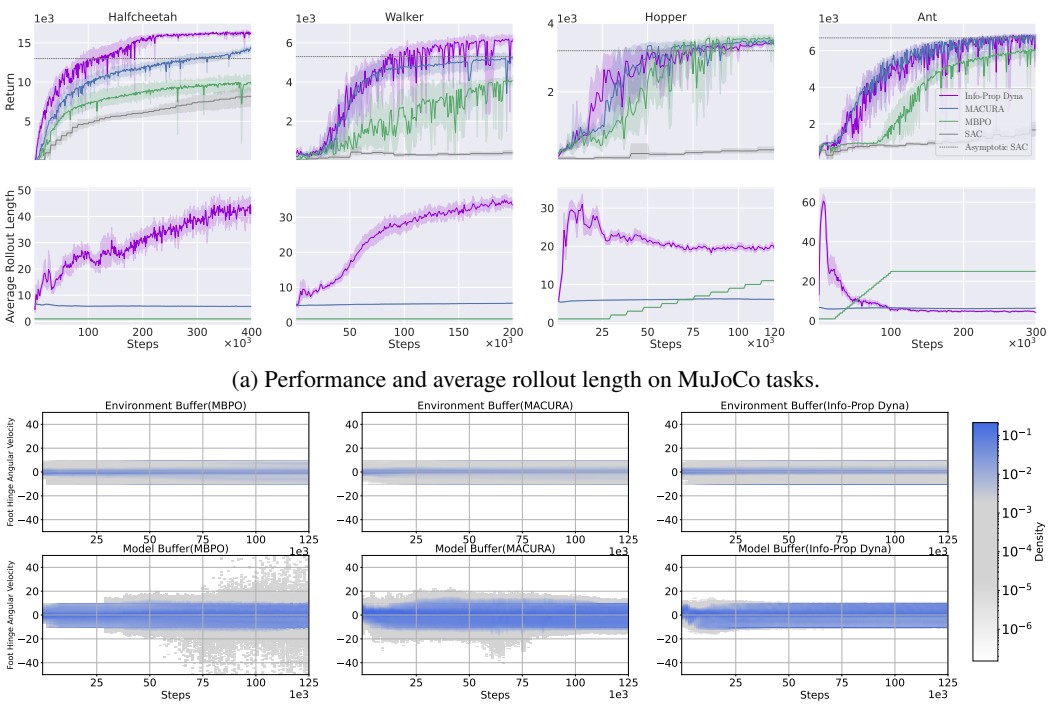

(a) Performance and average rollout length on MuJoCo tasks.

(b) Adequacy of $\mathcal{D}_{mod}$ on the $11^{th}$ state dimension of hopper.

Figure 5: Evaluation on MuJoCo tasks. *(a) Infoprop-Dyna shows state-of-the-art performance for Dyna-style MBRL on several MuJoCo tasks while considerably increasing average rollout length on most tasks. (b) Infoprop-Dyna shows substantially improved consistency between $\mathcal{D}_{env}$ and $\mathcal{D}_{mod}$.*

## 6.3 PERFORMANCE EVALUATION

As depicted in the top row of Figure 5a, Infoprop-Dyna performs on par with or better than MACURA, while substantially outperforming MBPO with respect to data efficiency and asymptotic performance. Notably, Infoprop-Dyna consistently outperforms SAC with a fraction of environment interaction. The bottom row of Figure 5a depicts the average rollout lengths. Infoprop-Dyna shows substantially increased rollout lengths compared to prior methods in all environments but ant.

A major concern of this work is the consistency of model-based rollouts with the environment distribution. Figure 5b depicts the data distribution in $\mathcal{D}_{env}$ and $\mathcal{D}_{mod}$ of the respective Dyna-style approaches throughout training for the $11^{th}$ dimension of the hopper state. The distributions are illustrated via histograms over environment steps. It can be seen that the model data distribution of Infoprop-Dyna closely follows the distribution observed in the environment, while both the data from MBPO and MACURA show severe outliers. This is the case, even though the rollout data in Infoprop-Dyna is obtained from substantially longer rollouts as can be seen from Figure 5a which indicates the capabilities of the Infoprop rollout mechanism.

### 6.4 LIMITATIONS AND OUTLOOK

Despite the excellent quality of model-generated data with the Infoprop rollout, the limitations of Infoprop-Dyna are most apparent on MuJoCo humanoid with results provided in Appendix E.3. These show instabilities in learning and point to structural problems when integrating Infoprop rollouts naively into standard Dyna-style architectures Janner et al. (2019).

Figure 5b shows that the long rollouts of Infoprop-Dyna can cause rapid distribution shifts in $\mathcal{D}_{\text{mod}}$, especially early in training. These nonstationary buffers are a well-known challenge to deep Q-learning methods Mnih et al. (2015). Another issue is primacy bias in model learning Qiao et al. (2023), where the model overfits to initial data and subsequently struggles to generalize, as seen in the decreasing rollout length for ant in Figure 5a. The main problem with Infoprop-Dyna is likely overfitting critics and plasticity loss Nikishin et al. (2022); D'Oro et al. (2023), as also reported by Frauenknecht et al. (2024) for Dyna-style MBRL trained on high-quality data. We provide an ablation on this observation and sketch methods to counteract this phenomenon in Appendix E.4.

## 7 RELATED WORK

The negative effects of accumulated model error on the performance of MBRL methods is a long-studied problem Venkatraman et al. (2015); Talvitie (2016); Asadi et al. (2018b;a).

Different model architectures have been proposed to mitigate this issue, such as trajectory models Asadi et al. (2019); Lambert et al. (2021), bidirectional models Lai et al. (2020), temporal segment models Mishra et al. (2017) or self-correcting models Talvitie (2016). These architectures, however, imply substantial additional effort for model learning, such that state-of-the-art performance in the respective fields of MBRL is often reported for simpler single-step model architectures Chua et al. (2018); Janner et al. (2019); Buckman et al. (2018).

These approaches address the problem of error accumulation by keeping model-based rollouts sufficiently short. Janner et al. (2019) introduce the concept of branched rollouts that allows to cover relevant parts of $\mathcal{S}$ with short model rollouts. Other methods weight rollouts of different lengths according to their single-step uncertainty Buckman et al. (2018) or use single-step uncertainty to schedule rollout length Pan et al. (2020); Frauenknecht et al. (2024). Infoprop allows to infer model data consistent with the environment distribution over long rollout horizons using comparatively simple model architectures and computationally cheap conditioning operations.

Infoprop is inspired by an information-theoretic view on RL Lu et al. (2023). Thus far, information-theoretic arguments have been mostly used to improve the exploration Haarnoja et al. (2018); Lu & Roy (2019); Ahmed et al. (2019); Mohamed & Rezende (2015) and generalization Tishby & Zaslavsky (2015); Lu et al. (2020); Igl et al. (2019); Islam et al. (2023) of model-free RL methods. While aspects of dynamical systems such as causality, modeling, and control Lozano-Duran & Arranz (2021), predictability Kleeman (2011) or dealing with noisy observations Gattami (2014) have been studied from an information theoretic perspective, these works do not directly apply to the MBRL setup nor extend to long model-based rollouts.

## 8 CONCLUDING REMARKS

Data consistency of model-based rollouts is a key criterion for the performance of MBRL approaches. This work proposes the novel Infoprop mechanism that substantially improves rollouts with common AES models. We reduce the influence of epistemic uncertainty on the predictive distribution of model-based rollouts, keep track of data corruption through propagated model error over long horizons, and terminate rollouts based on data corruption. This allows for considerably increased rollout lengths while substantially improving data consistency simultaneously.

While Infoprop is applicable to a broad range of MBRL methods, we demonstrate its capabilities by naively integrating Infoprop into a standard Dyna-style MBRL architecture Janner et al. (2019) resulting in the Infoprop-Dyna algorithm. We report state-of-the-art performance in several MuJoCo tasks while pointing to necessary adaptions to the existing algorithmic framework to fully unleash the potential of Infoprop rollouts.

ACKNOWLEDGMENTS

We thank Christian Fiedler, Pierre-François Massiani, and David Stenger for the fruitful discussions on the work presented in this paper. This work is funded in part under the Excellence Strategy of the Federal Government and the Länder (G:(DE-82)EXS-SF-OPSF854) and the German Federal Ministry of Education and Research (BMBF) under the Robotics Institute Germany (RIG), which the authors gratefully acknowledge. Friedrich Solowjow is supported by the KI-Starter grant by the state of NRW. Further, the authors gratefully acknowledge the computing time provided to them at the NHR Center NHR4CES at RWTH Aachen University (project number p0022301). This is funded by the Federal Ministry of Education and Research, and the state governments participating on the basis of the resolutions of the GWK for national high performance computing at universities (www.nhr-verein.de/unsere-partner).

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

# A   NOTATION

## A.1   RANDOM VARIABLES

$S_t$    Random variable of a general state
$\check{S}_t$    Random variable of the environment state
$\hat{S}_t$    Random variable of the model state
$\bar{S}_t$    Random variable of the estimated environment state
$\tilde{S}_t$    Random variable of the Infoprop state
$A_t$    Random variable of the action
$\Delta_t$    Random variable of the model error
$W_t$    Random variable of the aleatoric noise
$N_t$    Random variable of the epistemic noise
$U_t$    Random variable of the conditional noise
$\Theta_t$    Random variable of the model parameters

## A.2   REALIZATIONS

$s_t$    Realization of a general state
$\check{s}_t$    Realization of the environment state
$\hat{s}_t$    Realization of the model state
$\bar{s}_t$    Realization of the estimated environment state
$\tilde{s}_t$    Realization of the Infoprop state
$a_t$    Realization of the action
$w_t$    Realization of the aleatoric noise
$n_t$    Realization of the epistemic noise
$u_t$    Realization of the conditional noise
$\theta_t$    Realization of the model parameters

## A.3   TRANSITION KERNELS

$P_{\mathcal{S}}\left(\cdot|S_t, A_t\right) = \mathcal{N}\left(\mu(S_t, A_t), \Sigma(S_t, A_t)\right)$    Environment Transition Kernel
$\hat{P}_{\mathcal{S},\mathrm{TS}}(\cdot|S_t, A_t) = \mathcal{N}\left(\hat{\mu}_{\Theta_t}(S_t, A_t), \hat{\Sigma}_{\Theta_t}(S_t, A_t)\right)$    Trajectory Sampling Kernel
$\tilde{P}_{\mathcal{S},\mathrm{IP}}(\cdot|S_t, A_t, \hat{S}_{t+1}) = \mathcal{N}\left(\tilde{\mu}(S_t, A_t, \hat{S}_{t+1}), \tilde{\Sigma}(S_t, A_t, \hat{S}_{t+1})\right)$    Infoprop Kernel

## B  TOY EXAMPLE

In Figure 1, we illustrate the data consistency of Trajectory Sampling Chua et al. (2018) and Infoprop in a one-dimensional random walk example with $\mathcal{S} \subseteq \mathbb{R}$ and $\mathcal{A} \subseteq \mathbb{R}$. The dynamics follow (3) with $\mu(S_t, A_t) = S_t + A_t$ and $L(S_t, A_t) = 0.01$. Actions are distributed according to $A_t \sim \mathcal{N}(0, 0.1)$. All rollouts start from $s_0 = 0$ and are propagated for 100 steps. We perform 1000 rollouts under the environment dynamics and train a Probabilistic Ensemble Lakshminarayanan et al. (2017) model according to the information provided in Table 1. Subsequently, we perform 1000 model-based rollouts with this model and the respective rollout mechanism.

| Hyperparameter | Value |
|:---:|:---:|
| number of ensemble members | 5 |
| number of hidden neurons | 2 |
| number of layers | 1 |
| learning rate | 0.001 |
| weight decay | 0.00001 |
| number of epochs | 4 |

Table 1: Hyperparameters used for training the model on the random walk dataset.

## C  PSEUDOCODE ALGORITHMS

---
**Algorithm 2** Trajectory Sampling Chua et al. (2018)

---
**Require:** $s_0$
    **while** $t < T + 1$ **do**
        $a_t \sim \pi(\cdot | s_t)$
        $\hat{s}_{t+1} = \mathbb{E}\left[\hat{S}_{t+1} | W_t = w_t, \Theta_t = \theta_t\right]$ with $w_t \sim \mathcal{N}(0, I)$ and $\theta_t \sim \mathbb{P}_\Theta$
        $s_t \leftarrow \hat{s}_{t+1}$

---

---
**Algorithm 3** Infoprop-Dyna (Pseudocode adapted from Janner et al. (2019))

---
**Require:** Policy $\pi$, predictive AES model $p_\Theta$, environment buffer $\mathcal{D}_{\text{env}}$, model buffer $\mathcal{D}_{\text{mod}}$, rollout
    parameters $T, \zeta_1, \zeta_2, \xi$
    **for** $N$ epochs **do**
        **for** $J$ steps **do**
            Interact with the environment according to $\pi$; add to $\mathcal{D}_{\text{env}}$
        Train model $p_\Theta$ on $\mathcal{D}_{\text{env}}$
        Perform single-step predictions with $p_\Theta$ in $\mathcal{D}_{\text{env}}$
        Compute $\lambda_1$ (22) and $\lambda_2$ (23)
        **for** $M$ model rollouts **do**
            Sample $s_0$ uniformly from $\mathcal{D}_{\text{env}}$
            Perform Infoprop rollouts according to Algorithm 1; add to $\mathcal{D}_{\text{mod}}$
        **for** $G \cdot J$ gradient updates **do**
            Update $\pi$ on $\mathcal{D}_{\text{env}} \cup \mathcal{D}_{\text{mod}}$

---

# D DERIVATIONS

## D.1 QUANTIZED ENTROPY

For a RV $Z \in \mathcal{Z} \subseteq \mathbb{R}^{n_Z}$ with $Z \sim \mathcal{N}(\mu_Z, \Sigma_Z)$ and discretization step size $\Delta z^{(k)}$ of the $k^{\text{th}}$ dimension, the quantized entropy Cover & Thomas (2006) is

$$\mathbb{H}(Z) = \frac{1}{2} \log_2 \left( (2\pi e)^{n_Z} |\Sigma_Z| \right) - \sum_{k=1}^{n_Z} \log_2 \left( \Delta z^{(k)} \right). \tag{24}$$

## D.2 MAXIMUM LIKELIHOOD PREDICTIVE DISTRIBUTION

### D.2.1 PROOF OF LEMMA 1

*Proof.* We introduce the conditional expectation over the next state under the model, given a realization $\theta_t^e$

$$\hat{S}_{t+1}^e := \mathbb{E}_{\hat{P}_{\mathcal{S},\text{TS}}} \left[ \hat{S}_{t+1} | \Theta_t = \theta_t^e \right]. \tag{25}$$

Further, $\hat{\mu}^e := \hat{\mu}_{\Theta_t = \theta_t^e}$, $\hat{\Sigma}^e := \hat{\Sigma}_{\Theta_t = \theta_t^e}$ and $\hat{L}^e := \hat{L}_{\Theta_t = \theta_t^e}$ such that

$$\hat{S}_{t+1}^e = \hat{\mu}^e(S_t, A_t) + \hat{L}^e(S_t, A_t) W_t. \tag{26}$$

Given $E$ RVs $\hat{S}_{t+1}^e$ we define their joint distribution

$$\begin{pmatrix} \hat{S}_{t+1}^1 \\ \vdots \\ \hat{S}_{t+1}^E \end{pmatrix} \sim \mathcal{N} \left( \begin{bmatrix} \hat{\mu}^1 \\ \vdots \\ \hat{\mu}^E \end{bmatrix}, \begin{bmatrix} \hat{\Sigma}^1 & \cdots & \hat{\Sigma}^{1E} \\ \vdots & \ddots & \vdots \\ \hat{\Sigma}^{E1} & \cdots & \hat{\Sigma}^E \end{bmatrix} \right)$$

$$=: \quad \hat{S} \sim \mathcal{N} \left( \hat{\mu}, \hat{\Sigma} \right) \tag{27}$$

with $\hat{\Sigma}^{ef} := \text{Cov} \left[ \hat{S}_{t+1}^e, \hat{S}_{t+1}^f \right]$. We aim to track $S_{t+1}$ such that

$$H S_{t+1} \sim \mathcal{N} \left( \hat{\mu}, \hat{\Sigma} \right) \tag{28}$$

where we use $H = [I, I, \ldots, I]^\top \in \mathbb{R}^{n_{\mathcal{S}} \cdot E \times n_{\mathcal{S}}}$ to project $S_{t+1}$ to the dimension of the joint $\hat{S}$.

We define the maximum likelihood loss

$$\mathcal{L}(S_{t+1}) = p(\hat{S}|S_{t+1}) = \frac{1}{|2\pi\hat{\Sigma}|^{\frac{1}{2}}} \exp \left( -\frac{1}{2} \left( \hat{S} - H S_{t+1} \right) \hat{\Sigma}^{-1} \left( \hat{S} - H S_{t+1} \right) \right) \tag{29}$$

such that

$$\log \left( \mathcal{L}(S_{t+1}) \right) = -\frac{1}{2} \log \left( |2\pi\hat{\Sigma}| \right) - \frac{1}{2} \left( \hat{S} - H S_{t+1} \right) \hat{\Sigma}^{-1} \left( \hat{S} - H S_{t+1} \right). \tag{30}$$

We aim to obtain the maximizer of the log-likelihood such that

$$\bar{S}_{t+1} = \arg \max_{S_{t+1}} \log \left( \mathcal{L}(S_{t+1}) \right). \tag{31}$$

Consequently,

$$\frac{\partial}{\partial S_{t+1}} \log \left( \mathcal{L}(S_{t+1}) \right) = -\frac{1}{2} H^\top \hat{\Sigma}^{-1} \left( \hat{S} - H S_{t+1} \right) := 0$$

$$\Rightarrow \quad H^\top \hat{\Sigma}^{-1} \hat{S} - H^\top \hat{\Sigma}^{-1} H \bar{S}_{t+1} = 0 \tag{32}$$

$$\Rightarrow \quad \bar{S}_{t+1} = \left( H^\top \hat{\Sigma}^{-1} H \right)^{-1} H^\top \hat{\Sigma}^{-1} \hat{S}.$$

As a result, we obtain

$$\bar{\mu} = \mathbb{E} \left[ \bar{S}_{t+1} \right] = \left( H^\top \hat{\Sigma}^{-1} H \right)^{-1} H^\top \hat{\Sigma}^{-1} \hat{\mu} \tag{33}$$

and

$$
\begin{aligned}
\bar{\Sigma} = \mathrm{Var}\left[\bar{S}_{t+1}\right] &= \mathrm{Var}\left[\left(H^\top \hat{\Sigma}^{-1} H\right)^{-1} H^\top \hat{\Sigma}^{-1} \hat{S}\right] \\
&= \left(H^\top \hat{\Sigma}^{-1} H\right)^{-1} H^\top \hat{\Sigma}^{-1} \mathrm{Var}\left[\hat{S}\right] \hat{\Sigma}^{-1} H \left(H^\top \hat{\Sigma}^{-1} H\right)^{-1} \\
&= \left(H^\top \hat{\Sigma}^{-1} H\right)^{-1} H^\top \hat{\Sigma}^{-1} \hat{\Sigma} \hat{\Sigma}^{-1} H \left(H^\top \hat{\Sigma}^{-1} H\right)^{-1} = \left(H^\top \hat{\Sigma}^{-1} H\right)^{-1}
\end{aligned}
\tag{34}
$$

which corresponds to standard results in Kalman fusion. However, as the cross-correlations $\hat{\Sigma}^{ef}$ are unknown in practice, we approximate the Kalman fusion results (33) and (34) using covariance intersection fusion Julier & Uhlmann (2001) with uniform weights, making use of Assumption 1. This results in

$$
\bar{\Sigma} = \left(\frac{1}{E} \sum_{e=1}^{E} \left(\hat{\Sigma}^e\right)^{-1}\right)^{-1}
\tag{35}
$$

and

$$
\bar{\mu} = \bar{\Sigma} \left(\frac{1}{E} \sum_{e=1}^{E} \left(\hat{\Sigma}^e\right)^{-1} \hat{\mu}^e\right).
\tag{36}
$$

Hence, we can estimate the environment state as

$$
\bar{S}_{t+1} = \bar{\mu}(S_t, A_t) + \bar{L}(S_t, A_t) W_t,
\tag{37}
$$

with $\bar{L}\bar{L}^\top = \bar{\Sigma}$ and $W_t \sim \mathcal{N}(0, I)$. $\qquad\square$

### D.2.2 PROOF OF LEMMA 2

*Proof.* We continue here using the quantities we estimated in the previous section. To estimate $\Sigma^\Delta$, we interpret $\{\hat{\mu}^1\}_{e=1}^{E}$ as samples from a distribution whose mean is known to be $\bar{\mu}$. With this, the maximum likelihood estimate of $\Sigma^\Delta$ can be obtained trivially as

$$
\bar{\Sigma}^\Delta = \frac{1}{E} \sum_{e=1}^{E} \left(\hat{\mu}^e - \bar{\mu}\right)\left(\hat{\mu}^e - \bar{\mu}\right)^\top.
\tag{38}
$$

$\qquad\square$

### D.3 INFOPROP STATE

As introduced in (15), the Infoprop state is defined as

$$
\tilde{S}_{t+1} := \mathbb{E}\left[\bar{S}_{t+1} | \hat{S}_{t+1} = \hat{s}_{t+1}\right] = \mathbb{E}_{\tilde{P}_{\mathcal{S},\mathrm{IP}}}\left[S_{t+1} | S_t = s_t, A_t = a_t, \hat{S}_{t+1} = \hat{s}_{t+1}, U_t\right]
\tag{39}
$$

Combining (11) and Assumption 2, we have

$$
\hat{S}_{t+1} = \check{S}_{t+1} + L^\Delta(S_t, A_t) N_t.
\tag{40}
$$

Plugging the respective maximum likelihood estimates into (40) yields

$$
\hat{S}_{t+1} = \bar{S}_{t+1} + \bar{L}^\Delta(S_t, A_t) N_t
\tag{41}
$$

with

$$
\bar{S}_{t+1} = \bar{\mu}(S_t, A_t) + \bar{L}(S_t, A_t) W_t
\tag{42}
$$

according to (13). As we can generally consider model uncertainty as independent from process noise, i.e. $N_t \perp W_t$, the Infoprop state $\tilde{S}_{t+1} = \mathbb{E}[\check{S}_{t+1} | \hat{S}_{t+1} = \hat{s}_{t+1}]$ can be computed using a standard Kalman update.

The general form of the Kalman update Simon (2006) considers two Gaussian RVs $X \sim \mathcal{N}(\mu_X, \Sigma_X)$ and $Y = X + N$ with $N \sim \mathcal{N}(0, \Sigma_N)$ and $X \perp N$. Then, given an observation $y$ we can compute the conditional expectation of $X$

$$
\mathbb{E}\left[X | Y = y\right] \sim \mathcal{N}\left(\mu_{X|Y=y}, \Sigma_{X|Y=y}\right)
\tag{43}
$$

with

$$\mu_{X|Y=y} = \mu_X + K(y - \mu_X), \tag{44}$$

$$\Sigma_{X|Y=y} = (I - K)\,\Sigma_X, \tag{45}$$

and

$$K = \Sigma_X \left(\Sigma_X + \Sigma_N\right)^{-1}. \tag{46}$$

Following (15), we can compute the Infoprop state via (43) choosing

$$\mu_X = \bar{\mu}(s_t, a_t), \tag{47}$$

$$\Sigma_X = \bar{\Sigma}(s_t, a_t), \tag{48}$$

$$\Sigma_N = \bar{\Sigma}^\Delta(s_t, a_t), \tag{49}$$

and

$$y = \bar{\mu}(s_t, a_t) + \bar{L}(s_t, a_t)w_t + \bar{L}^\Delta(s_t, a_t)n_t. \tag{50}$$

This yields the propagation equation of the Infoprop state

$$\tilde{S}_{t+1} = \tilde{\mu}(S_t = s_t, A_t = a_t, \hat{S}_{t+1} = \hat{s}_{t+1}) + \tilde{L}(S_t = s_t, A_t = a_t, \hat{S}_{t+1} = \hat{s}_{t+1})U_t \tag{51}$$

with

$$\tilde{\mu}(s_t, a_t, \hat{s}_{t+1}) = \bar{\mu}(s_t, a_t) + K(s_t, a_t)\left(\bar{L}(s_t, a_t)w_t + \bar{L}^\Delta(s_t, a_t)n_t\right), \tag{52}$$

$$\tilde{\Sigma}(s_t, a_t, \hat{s}_{t+1}) = (I - K(s_t, a_t))\,\bar{\Sigma}(s_t, a_t), \tag{53}$$

$$K(s_t, a_t) = \bar{\Sigma}(s_t, a_t)\left(\bar{\Sigma}(s_t, a_t) + \bar{\Sigma}^\Delta(s_t, a_t)\right)^{-1}, \tag{54}$$

$$\tilde{L}(s_t, a_t)\tilde{L}(s_t, a_t)^\top = \tilde{\Sigma}(s_t, a_t), \tag{55}$$

and

$$\bar{L}^\Delta(s_t, a_t)\bar{L}^\Delta(s_t, a_t)^\top = \bar{\Sigma}^\Delta(s_t, a_t). \tag{56}$$

### D.4 Induced State Distribution by the Infoprop Rollout

**Lemma 3.** *As introduced in (57), the next state distribution induced by the Infoprop rollout is the same as that given by the estimated ground truth:*

$$\tilde{S}_{t+1} \overset{\text{dist}}{=} \bar{S}_{t+1} \tag{57}$$

*Proof.* We show equality in distribution via comparison of the cumulative distribution functions (CDF) of $\tilde{S}_{t+1}$ and $\bar{S}_{t+1}$. If we can show that the CDFs are identical, i.e. $\mathbb{P}(\tilde{S}_{t+1} \leq \bar{s}_{t+1}) = \mathbb{P}(\bar{S}_{t+1} \leq \bar{s}_{t+1}) \quad \forall \bar{s}_{t+1} \in \mathcal{S}$, the equality in distribution follows.

We compute $\mathbb{P}(\tilde{S}_{t+1} \leq \bar{s}_{t+1})$ using $\tilde{S}_{t+1} = \mathbb{E}[\bar{S}_{t+1}|\hat{S}_{t+1} = \hat{s}_{t+1}]$ and marginalizing over $\hat{S}_{t+1}$

$$\mathbb{P}(\tilde{S}_{t+1} \leq \bar{s}_{t+1}) = \int_{\mathcal{S}} \mathbb{P}(\mathbb{E}[\bar{S}_{t+1}|\hat{S}_{t+1}] \leq \bar{s}_{t+1}|\hat{S}_{t+1} = \hat{s}_{t+1})f_{\hat{S}_{t+1}}(\hat{s}_{t+1})\mathrm{d}\hat{s}_{t+1} \tag{58}$$

with $f_{\hat{S}_{t+1}}$ the probability density function of $\hat{S}_{t+1}$.

By construction, $\mathbb{E}[\bar{S}_{t+1}|\hat{S}_{t+1}]$ describes the behavior of $\bar{S}_{t+1}$ given $\hat{S}_{t+1}$. Consequently,

$$\mathbb{P}(\mathbb{E}[\bar{S}_{t+1}|\hat{S}_{t+1}] \leq \bar{s}_{t+1}|\hat{S}_{t+1} = \hat{s}_{t+1}) = \mathbb{P}(\bar{S}_{t+1} \leq \bar{s}_{t+1}|\hat{S}_{t+1} = \hat{s}_{t+1}) \tag{59}$$

and therefore

$$\mathbb{P}(\tilde{S}_{t+1} \leq \bar{s}_{t+1}) = \int_{\mathcal{S}} \mathbb{P}(\bar{S}_{t+1} \leq \bar{s}_{t+1}|\hat{S}_{t+1} = \hat{s}_{t+1})f_{\hat{S}_{t+1}}(\hat{s}_{t+1})\mathrm{d}\hat{s}_{t+1}. \tag{60}$$

The right hand side of (60) represents the law of total probability for $P(\bar{S}_{t+1} \leq \bar{s}_{t+1})$

$$\mathbb{P}(\bar{S}_{t+1} \leq \bar{s}_{t+1}) = \int_{\mathcal{S}} \mathbb{P}(\bar{S}_{t+1} \leq \bar{s}_{t+1}|\hat{S}_{t+1} = \hat{s}_{t+1})f_{\hat{S}_{t+1}}(\hat{s}_{t+1})\mathrm{d}\hat{s}_{t+1}. \tag{61}$$

Therefore, we have

$$\mathbb{P}(\tilde{S}_{t+1} \leq \bar{s}_{t+1}) = \mathbb{P}(\bar{S}_{t+1} \leq \bar{s}_{t+1}) \quad \forall \bar{s}_{t+1} \in \mathcal{S} \tag{62}$$

and can conclude

$$\tilde{S}_{t+1} \overset{\text{dist}}{=} \bar{S}_{t+1}. \tag{63}$$

$\square$

### D.5 INFORMATION LOSS ALONG A INFOPROP ROLLOUT

**Lemma 4.** *As introduced in (17), the total information loss incurred during a Infoprop equals the accumulated entropy of the Infoprop state:*

$$\mathbb{H}\left(\bar{S}_1, \bar{S}_2, \ldots, \bar{S}_T | S_0 = s_0, A_0 = a_0, \hat{S}_1 = \hat{s}_1 \ldots \hat{S}_T = \hat{s}_T\right) = \sum_{t=0}^{T-1} \mathbb{H}\left(\tilde{S}_{t+1}\right) \qquad (64)$$

*Proof.*

$$
\begin{aligned}
&\mathbb{H}\left(\bar{S}_1, \bar{S}_2, \ldots, \bar{S}_T | S_0 = s_0, A_0 = a_0, \hat{S}_1 = \hat{s}_1 \ldots \hat{S}_T = \hat{s}_T\right) \\
&= \sum_{t=0}^{T-1} \mathbb{H}\left(\bar{S}_{t+1} \mid \bar{S}_1, \bar{S}_2, \ldots, \bar{S}_t, S_0 = s_0, A_0 = a_0, \hat{S}_1 = \hat{s}_1 \ldots \hat{S}_T = \hat{s}_T\right) \\
&\stackrel{(a)}{=} \sum_{t=0}^{T-1} \mathbb{H}\left(\bar{S}_{t+1} \mid \bar{S}_1, \bar{S}_2 \ldots \bar{S}_t, S_0 = s_0, A_0 = a_0, \hat{S}_1 = \hat{s}_1, \ldots, \hat{S}_T = \hat{s}_t\right) \\
&\stackrel{(b)}{=} \sum_{t=0}^{T-1} \mathbb{H}\left(\bar{S}_{t+1} \mid S_t = s_t, A_t = a_t, \hat{S}_{t+1} = \hat{s}_{t+1}\right) \\
&= \sum_{t=0}^{T-1} \mathbb{H}\left(\tilde{S}_{t+1}\right)
\end{aligned}
\qquad (65)
$$

where (a) follows from causality and (b) follows from the Markov property. $\qquad \square$

# E   EXPERIMENTS

## E.1   EXPERIMENTAL SETUP

We used Weights&Biases [4] for logging our experiments and run 5 random seeds per experiment.

The respective hyperparameters for Infoprop-Dyna on MuJoCo are given below. Table 2 addresses model learning, Table 3 the Infoprop mechanism, and Table 4 training the model-free agent.

Table 2: Hyperparameters used to train the model of Infoprop-Dyna in the Mujoco Tasks.

| Hyperparameter | Halfcheetah | Walker | Hopper | Ant |
|---|---|---|---|---|
| ensemble size $E$ | 7 | | | |
| number of hidden neurons | 200 | | | 400 |
| number of hidden layers | 4 | | | |
| learning rate | 0.0003 | 0.0006 | 0.0004 | 0.001 |
| weight decay | 0.00005 | 0.0007 | 0.0008 | 0.00002 |
| patience for early-stopping | 10 | 9 | 8 | 9 |
| retrain interval | 250 environment steps | | | |

---

[4] https://wandb.ai/site

Table 3: Hyperparameters of the Infoprop rollouts in the Mujoco Tasks.

| Hyperparameter | Halfcheetah | Walker | Hopper | Ant |
|---|---|---|---|---|
| accurate quantile $\zeta_1$ | 0.99 | | | |
| exceptionally accurate quantile $\zeta_2$ | 0.01 | | | |
| scaling factor $\xi$ | 100 | | | |
| rollout interval | 250 environment steps | | | |
| rollout batch size | 100000 | | | |

Table 4: Hyperparameters used to train the SAC agent of Infoprop-Dyna in the Mujoco Tasks.

| Hyperparameter | Halfcheetah | Walker | Hopper | Ant |
|---|---|---|---|---|
| number of hidden neurons | 1024 | | 512 | 1024 |
| number of hidden layers | 2 | | | |
| learning rate | 0.0003 | 0.0002 | 0.0004 | 0.0005 |
| SAC target entropy | -6 | -7 | 1 | 0 |
| target update interval | 1 | 4 | 6 | 5 |
| update steps $G$ | 10 | | | 20 |

The results for SAC, MBPO and MACURA are obtained from Frauenknecht et al. (2024).

### E.2 PREDICTION QUALITY

We provide additional results for the rollout consistency experiments introduced in Section 6.2. Figure 6 depicts model-based rollouts for the $10^{\text{th}}$ dimension of hopper under MBPO, MACURA and Infoprop-Dyna when setting the maximum rollout length of all approaches to 100. In the original experiment depicted in Figure 4b the maximum rollout length was 11 for MBPO and 10 for MACURA, following the hyperparameter settings reported in the respective publications Janner et al. (2019); Frauenknecht et al. (2024).

We observe a vastly spread distribution of MBPO rollouts, as every rollout is propagated for 100 steps, irrespective of model uncertainty, as long as it does not reach a terminal state of the hopper task. MACURA rollouts have an improved consistency compared to MBPO, especially in the beginning of the rollouts. Over long horizons, however, the TS propagation mechanism and the single-step termination criterion cannot produce consistent data. In contrast, Infoprop-Dyna is able to propagate consistent rollouts over long horizons.

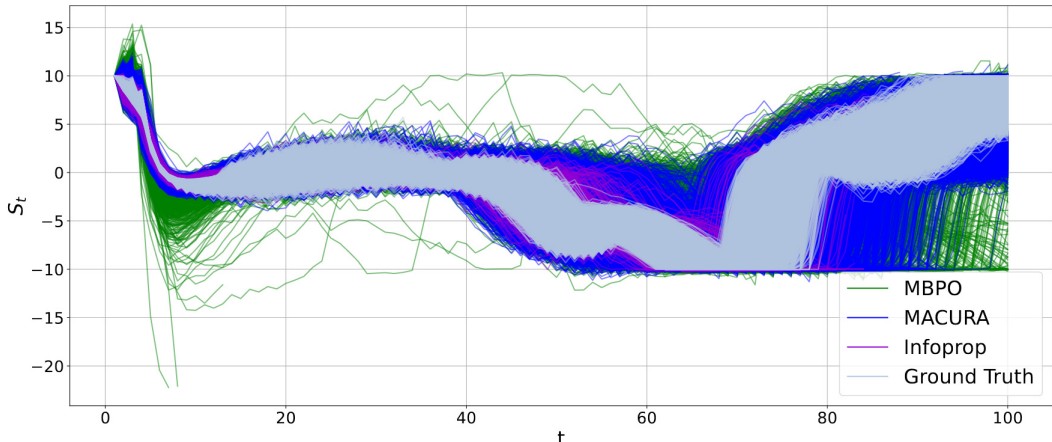

Figure 6: Rollout consistency MBPO vs. MACURA vs. Infoprop-Dyna for 100 steps. *Comparison of the respective rollout mechanisms similar to Figure 4b but with a maximum rollout length of 100 for all approaches.*

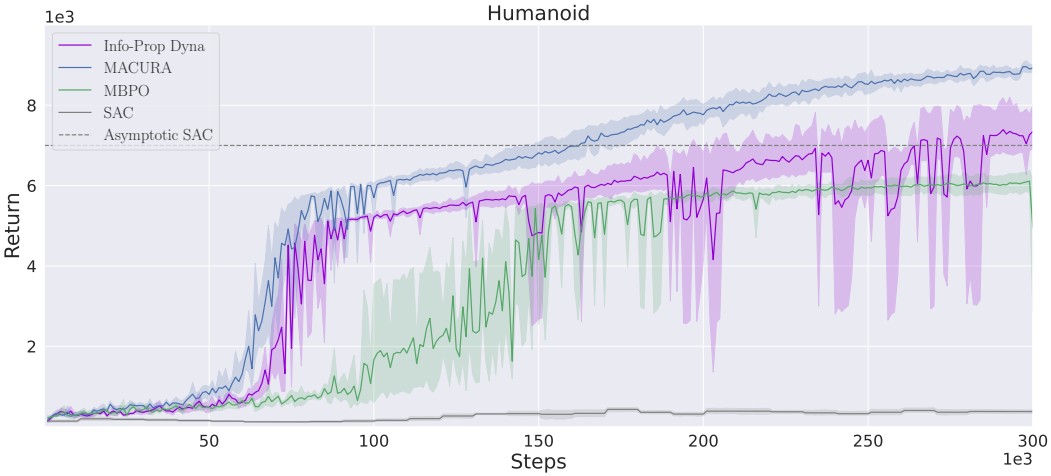

Figure 7: Performance on Humanoid

### E.3 PERFORMANCE ON HUMANOID

Figure 7 depicts the return on MuJoCo humanoid. We observe instabilities in the performance of Infoprop-Dyna towards the end of training. We assume this occurs due to overfitting and plasticity loss in the critic of the model-free learner Nikishin et al. (2022); D'Oro et al. (2023). This is reflected in the peaking critic loss depicted in Figure 8 concurrently with the performance drops. We set the update ratio $G$ (see Algorithm 3) to a relatively low value of 10 which explains the slower learning behavior than MACURA. For higher values of $G$, instabilities occur even earlier in the training process, underscoring our assumption of overfitting critics.

Model rollout inconsistency does not appear to be the destabilizing factor, as rollout data is consistent with the environment distribution as depicted in Figure 10 and the rollout adaption mechanism seems to react to policy shifts induced by high critic losses through reducing the average rollout length as depicted in Figure 10.

### E.4 INVESTIGATING INSTABILITIES IN LEARNING

Although Infoprop gives better quality data over longer rollout horizons than TS rollouts, we observe instabilities in learning when naively integrating Infoprop into the conventional Dyna setting. We hypothesize that the main cause of these instabilities is due to the agent overfitting to the higher

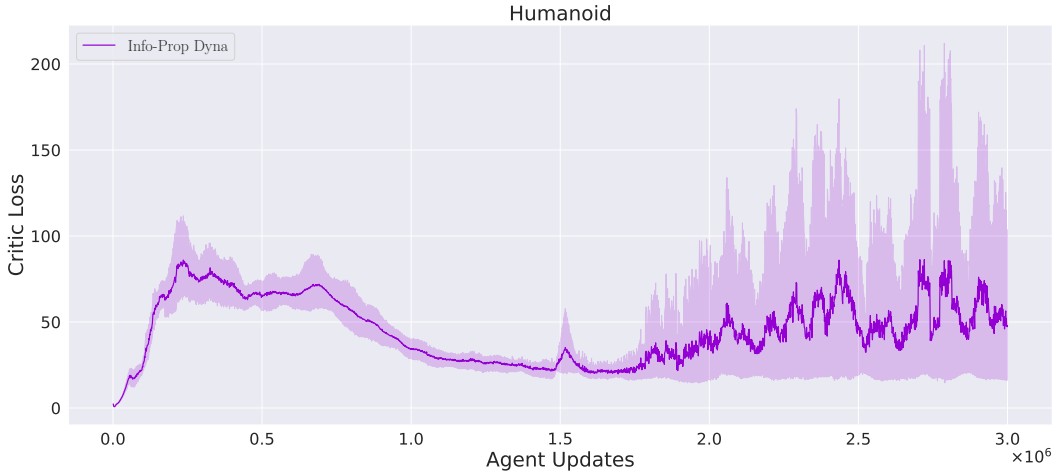

Figure 8: Critic Loss on Humanoid

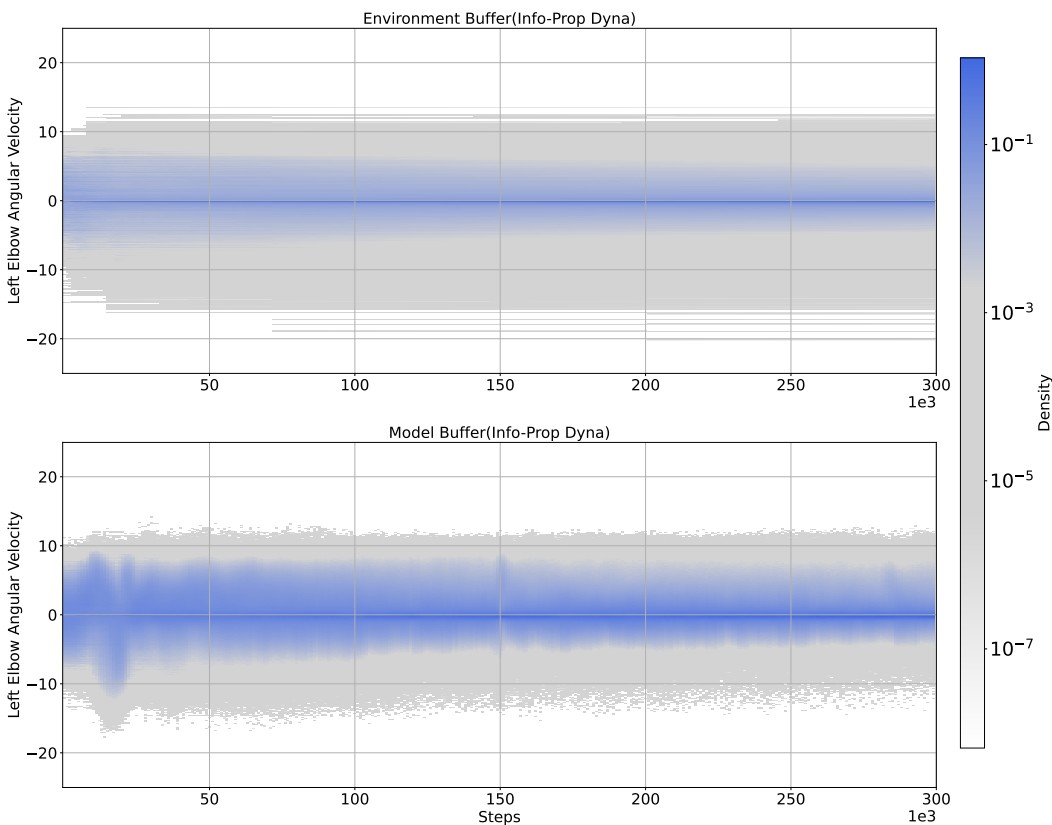

Figure 9: Comparison between $\mathcal{D}_{\text{env}}$ and $\mathcal{D}_{\text{mod}}$ for the $45^{\text{th}}$ dimension of Humanoid

quality data produced by Infoprop rollouts, followed by loss of plasticity Nikishin et al. (2022); D'Oro et al. (2023). To investigate this, we carried out an ablation by varying the values of $\zeta_1$, which we introduced in Equation 22. This hyperparameter controls the size of the subset $\mathcal{E}$ where the model is considered sufficiently accurate. The smaller the value of $\zeta_1$, the more aggressive the filtering of single-step information losses, leading to a smaller $\mathcal{E}$.

Figure 11 shows the returns obtained on the Hopper task for three values of $\zeta_1$. For $\zeta_1 = 0.97$, we see that the returns are unstable throughout training, even though this setting gives the best quality data. On the other hand, $\zeta_1 = 0.9999$ produces a more stable learning curve compared to

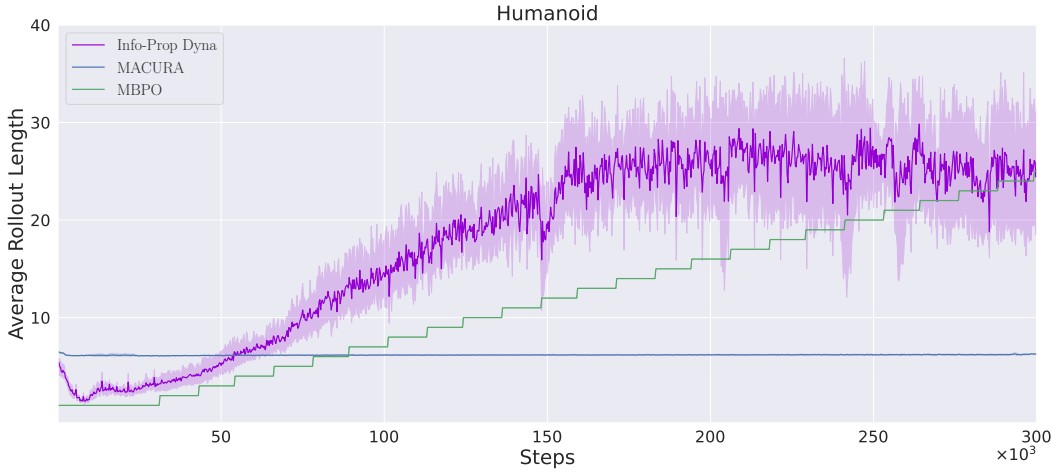

Figure 10: Average Rollout Length on Humanoid

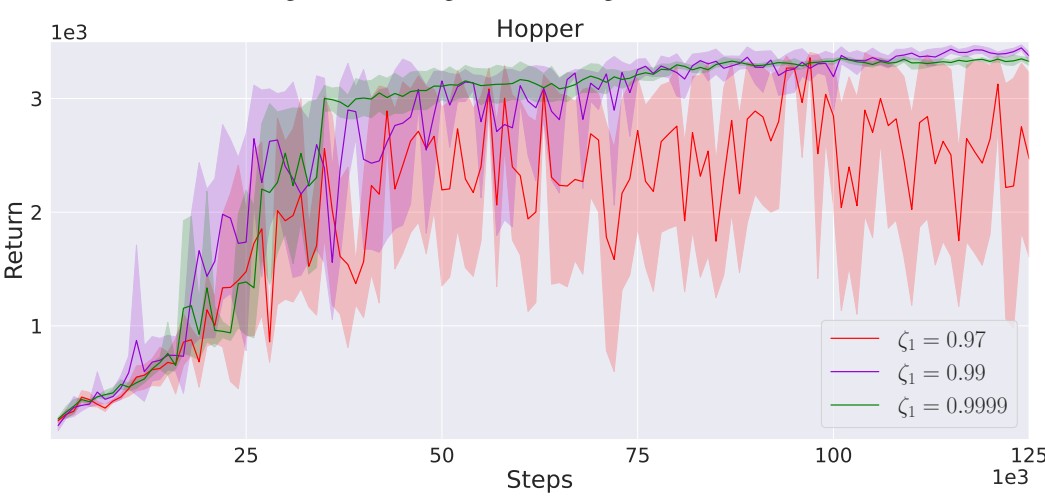

Figure 11: Ablation Study on Hopper.

$\zeta_1 = 0.99$, which was used for all the experiments in Section 6. This shows that better data quality does not necessarily lead to better training performance since if that was the case, $\zeta_1 = 0.97$ would have produced the best performance. A similar observation is reported in Frauenknecht et al. (2024), where low values of the scaling factor $\xi$, corresponding to accurate model rollouts, led to instabilities in learning.

Our observations show that producing high-quality synthetic data in the conventional Dyna setting leads to issues seen in MFRL when using a high update-to-data (UTD) ratio. There have been recent works on regularization methods to counteract agent overfitting and loss of plasticity. One such approach is applying layer normalization Smith et al. (2023); Nauman et al. (2024). Figure 12 shows the same settings as in Figure 11 but with layer normalization applied to the critic and actor networks. It can be seen that even for $\zeta_1 = 0.97$, the learning is stable.

The primary aim of this paper is to introduce the conceptual framework of the Infoprop rollout, as well as show its application to MBRL. Hence, we do not spend additional effort on tuning the hyperparameters or adding regularizations since this takes us away from the main objective. We defer such enhancements for future work.

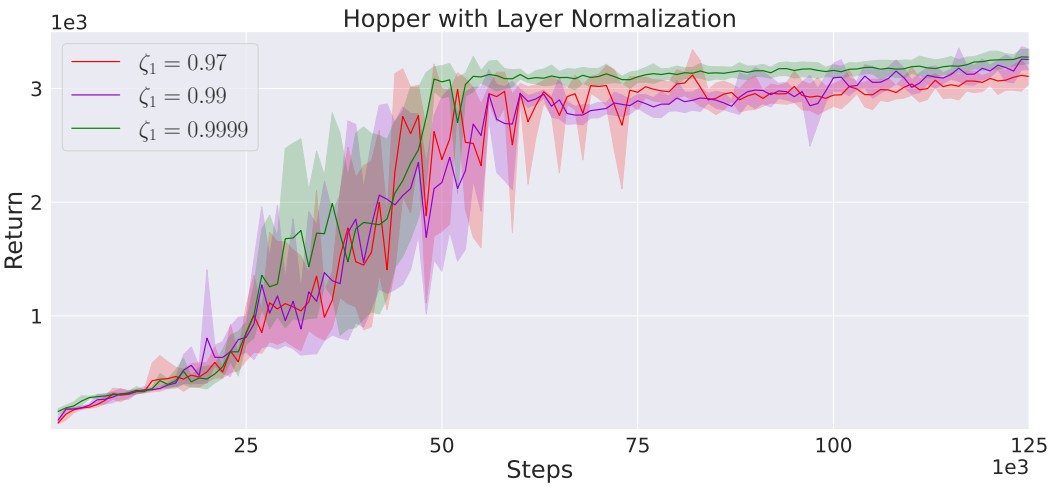

Figure 12: Ablation Study on Hopper with layer normalization.

