# OpenReview forum: "On Rollouts in Model-Based Reinforcement Learning"
_ICLR.cc/2025/Conference — ICLR 2025 Poster_

### Official Review · Reviewer_Anzx · 2024-10-16

**Soundness:** 4
**Presentation:** 3
**Contribution:** 3
**Rating:** 6
**Confidence:** 4

**Summary:**

The author presents a new method called *Infoprop*, which can be used as a plugin for Dyna-style model-based RL. The plugin increases the quality of data rollouts from the estimated model, thus enhancing the Dyna-style model-based RL. The experiment in Mujoco showed the SOTA performance except for the humanoid environment. The author also provides a solid theoretical analysis of why the plugin can increase the quality of rollouts based on the estimated model.

**Strengths:**

The author developed a new method, Infoprop, that enhances the performance of Dyna-style model-based RL:

*Versatility*: The Infoprop method can be integrated as a plugin for Dyna-style model-based RL, making it a flexible and adaptable enhancement for existing frameworks.

*Improved Data Quality*: By increasing the quality of data rollouts from the estimated model, Infoprop enhances the overall performance of Dyna-style RL approaches.

*Experimental Results*: The experiments conducted in Mujoco environments demonstrate state-of-the-art (SOTA) performance, showcasing the effectiveness of Infoprop in most environments, with the exception of the humanoid scenario.

*Theoretical Foundation*: The method is backed by robust theoretical analysis, explaining how the plugin improves rollout quality from the estimated model, which adds credibility to the approach.

**Weaknesses:**

1. The theoretical results in Section 4 are difficult to follow. A summary of the main result should be given at the start of the section. Also, I suggest the author clearly present the main result as a theorem.
2. Too many notations have been introduced, which could be simplified. For example, Section 2.4 could be condensed into a single sentence, accompanied by equation (5) and relevant citations since it represents a well-known formulation in model-based RL.
3. The enhancement applies specifically to Dyna-style RL, not all model-based RL approaches. Making the title misleading.
4. The Assumptions at the end of Section 4.2 are stated without any explainations.

**Questions:**

In Figure 6, INFOPROP Dyna does not outperform MACURA. However, if I understand correctly, the proposed method can be used as a plugin within MACURA, which makes it puzzling that INFOPROP Dyna does not surpass MACURA.

---

> ### Author Response · Authors · 2024-11-18
>
> **W1:** Thank you for pointing this out. We agree and have revised Section 4 accordingly, as discussed in point 1 of the general rebuttal.
>
> **W2:** Thank you for pointing this out. We strongly believe that all introduced notation is necessary to precisely distinguish the different rollout configurations which is crucial for a clear presentation of our results. We have put in a lot of effort to make our notation as compact as possible. However, we agree that there is a lot of notation one needs to keep track of. We plan to have a notation list that includes descriptions, definitions, and update rules of different objects in Appendix A. This Appendix can be opened on a second screen and can help navigate our notation. In order to upload the revised version in a timely manner, we have not finished this list yet but will do so for a potential camera-ready version.
>
> **W3:** The main contribution of the paper is the Infoprop rollout mechanism that is applicable to different types of MBRL algorithms using AES models. We illustrate the capabilities in a Dyna-style architecture, however, Infoprop is not limited to these architectures. Infoprop could benefit all MBRL algorithms using TS rollouts, e.g. the ones mentioned at the end of Section 2.4. these comprise model-based planning, analytic-gradient and Dyna-style architectures. Infoprop should likewise be applicable to model-based value expansion methods, e.g. a STEVE [1] -style approach using probabilistic ensemble models.
>
>       [1] Sample-Efficient Reinforcement Learning with Stochastic Ensemble Value Expansion, Buckman et al. 2018
>
> **W4:** Thank you for pointing this out, we have addressed this concern in point 2 of the general rebuttal.
>
> **Q1:** Thank you for your question. The reason we present Infoprop-Dyna is threefold: (1) to give an example on how to define $\lambda_1$ and $\lambda_2$ in practice, (2) to show that naively integrating Infoprop into a given algorithmic architecture performs reasonably well, and (3) to showcase the substantially improved quality of predictive data in a realistic scenario. We do not aim to propose an extremely high-performing Dyna-style algorithm. Thus, Infoprop-Dyna builds on the simpler MBPO architecture (e.g. we do not have an adaptive number of critic updates as MACURA). Further, we did not put additional effort into fine-tuning the algorithm or counteract the reported critic overfitting issues, as this would distract from the intended main contribution of the Infoprop rollout scheme. We have added an ablation discussing tuning and stabilizing the algorithm in Appendix E4. Consequently, we do not consider it surprising or critical that MACURA outperforms Infoprop-Dyna on humanoid.

---

> > ### Comment · Reviewer_Anzx · 2024-11-24
> >
> > Thanks for the response. The score remains.

---

### Official Review · Reviewer_uFDs · 2024-10-26

**Soundness:** 4
**Presentation:** 3
**Contribution:** 4
**Rating:** 8
**Confidence:** 3

**Summary:**

This paper proposes a model rollout method for model-based reinforcement learning called infoprop. The main idea is to control for epistemic uncertainty using model rollout to reduce model error accumulation and in turn improve value or policy optimization. The concrete method is a re-weighting of ensemble predictions and an entropy-based criteria to stop rollout. The Mujoco experiments demonstrated substantial improvement in reducing error accumulation as well as convergence speed.

**Strengths:**

The proposed method is original and sound. The experiment hypotheses are very clear the the results clearly demonstrated the advantage of the proposed method.

**Weaknesses:**

The paper could benefit a lot from intuitive explanations of the proposed approach. For example, what's the intuition of the covariance intersection fusion result for readers that are not familiar with this literature.

**Questions:**

* How does eq 14 actually follow from eq 4 and 9?
* When the authors say eliminate epistemic uncertainty, does that mean setting the last term in eq 9 to zero? If so is the rest of the derivations simply accounting for $\mu^{\Delta}$?

---

> ### Author Response · Authors · 2024-11-18
>
> **W1:** Thank you for pointing this out. As discussed in point 1 of the general rebuttal, we did a major revision on how Section 4 is presented. We now provide an intuitive explanation of the Infoprop approach itself in the beginning of Section 4. Concerning covariance intersection fusion, we rather consider this a technical detail that is not crucial for understanding the general approach. Thus, we have moved this to Appendix D2. If requested, we can provide a geometrical interpretation of CI fusion. For now, we have decided not to do so to keep the appendix as compact as possible.
>
> **Q1:** Thank you for the valuable remark. We now provide the full derivation of the Infoprop state in Appendix D3.
>
> **Q2:** The term 'eliminating epistemic uncertainty' was a misleading formulation. Pointing us to this helped improve the presentation of Section 4. What we meant is that we want to remove the influence of epistemic uncertainty from the predictive distribution. However, we cannot set N to 0 in practice as we don't have access to N directly. Instead, we have to sample parameter realizations and interpret them as epistemic noise realizations (see discussion around Definition 4). From these samples, we estimate the environment distribution and the noise distribution. Finally, we want to make sure that Infoprop follows the estimated environment distribution (which it does, see Theorem 1 (i)) while neglecting the noise distribution. We hope the presentation is clearer now.

---

> > ### Comment · Reviewer_uFDs · 2024-11-20
> >
> > I thank the authors for their excellent effort for the revision. The new version is substantially clearer. I have no further comments.

---

### Official Review · Reviewer_NHA5 · 2024-11-03

**Soundness:** 2
**Presentation:** 2
**Contribution:** 2
**Rating:** 5
**Confidence:** 3

**Summary:**

This paper presents a rollout mechanism that uses information theory to reduce the impact of epistemic uncertainty in MBRL. The paper presents some experimental analysis and a derivation of a result related to information loss.  The paper uses the idea of marginal entropy and mutual information to determine data corruption and stop a rollout.

**Strengths:**

Soundness
======
The paper uses the idea of marginal entropy and mutual information to determine data corruption and stop a rollout in MBRL. The idea is presented carefully with a derivation for the main result.

Significance & Related work
=========
The paper has a brief related work section, as part of the main paper, which is appreciated. However, there is a need for a deeper analysis of existing work, including  more in-depth comparisons that show the benefit of Infoprop to others, specifically including work in using simulated rollouts to guarantee performance, robust MBRL (Sung 2024, Kuo 2021), risk sensitivity (e.g. Webster 2021) and other uses of information theory in MBRL.

Presentation
=========
The paper is well written with few typos, usually around empty spaces in the text.

Quality
======
My main concerns are around the quality of the experimentation (see below) and on the need for authors to explore the limitations of their approach a bit deeper before publication. For example, the authors mention that the rollouts exhibit excellent data quality and yet there are instabilities: is this a function of the integration mechanism or of overfitting?

**Weaknesses:**

Experimentation
=========
The experimental analysis show the benefits of Infoprop when it runs in simple environments, however deeper experimentation is needed. For example, I would have expected to see some ablation studies showing the impact of the choice of lambda, instability, overfitting and other issues.

Overall the paper presents a very promising initial idea with several avenues of experimentation that need to be explored to ensure its maximum potential is reached.

**Questions:**

The main limitations of the work, as identified above, are in the quality and significance as discussed above. These could be alleviated with a more in-depth experimentation which would lead to a more mature solution, and a more in-depth discussion of related work. All of them require space, and a such my questions relate to the guidance for the discussion section.

Specifically:
1. where is overfitting most prominent with Infoprop?
2. where there other information theoretic metrics (e.g redundancy) considered in the Infoprop design? If no, why not? if yes, why were they discarded?

---

> ### Author Response · Authors · 2024-11-18
>
> **Related Work:** Thank you for the remark, we are happy to discuss how to incorporate these papers. However,  while the proposed works address interesting problems, we are not exactly sure how to incorporate them into the related work section. While these methods could potentially benefit from using Infoprop rollouts, e.g. being less conservative wrt. safety certification as the uncertainty can be specified more precisely, their key contributions are different from our approach. So from our understanding, these works are potential application fields of Infoprop but not primarily works we need to differentiate our approach from. Do you agree on this or did we get you wrong?
>
> Further, we are not aware of further literature regarding information theory in MBRL to discuss in the related work section. In case you have works in mind we missed, we would be happy to incorporate them.
>
> **W1:** Thank you for well thought through feedback. We have added an ablation study in Appendix E.4, investigating the instabilities in learning resulting from training the agent using high-quality data. We performed experiments on the Hopper task with three different values for the hyperparameter $\zeta_1$. The smaller the value of $\zeta_1$, the more aggressive the filtering of single-step information losses, resulting in fewer inaccurate model transitions being added to the replay buffer. We observe that the training destabilizes for the lower value of $\zeta_1=0.97$. In fact, the higher value $\zeta_1=0.9999$ gives a more stable learning performance.
>
> We hypothesize that using higher-quality data in MBRL causes the same destabilizing issues faced by model-free RL (MFRL) when using a high update-to-data (UTD) ratio[1][2]. To counteract this effect, different regularization approaches have been developed for MFRL, the most prominent of which is layer normalization[3][4]. We performed a separate set of experiments using layer normalization for the actor and critic networks and observed that this does stabilize training even for $\zeta_1=0.97$. Hence, in our opinion, the instabilities we report are not due to our proposed rollout scheme rather due to pre-existing failure cases with the model-free agent. Since the objective of this paper is to lay out the mathematical framework for Infoprop, and show its applicability to MBRL, we do not do extensive hyperparameter tuning or incorporate regularization in the main results.
>
>     [1] The primacy bias in deep reinforcement learning, Nikishin et al., 2022.
>
>     [2] Sample-efficient reinforcement learning by breaking the replay ratio barrier, D'Oro et al., 2023.
>
>     [3] Demonstrating a Walk in the Park: Learning to Walk in 20 Minutes With Model-Free Reinforcement Learning, Smith et al., 2023.
>
>     [4] Overestimation, Overfitting, and Plasticity in Actor-Critic: the Bitter Lesson of Reinforcement Learning, Nauman et al., 2024.
>
> **Q1:** As mentioned in W1, the overfitting is more prominent when using more aggressive filtering of model transitions, which gives better quality data. We have shown that regularization approaches developed for using high UTD in MFRL can be used to mitigate the instabilities in learning.
>
> **Q2:** Q3: There is a set of requirements for the metrics used for such a rollout:
> - The resulting RV needs to follow the environment distribution.
> - The resulting sequence of RVs needs to provide a notion of information loss.
> - This notion of information loss needs to allow for a feasible cutoff criterion (e.g. as the conditional entropy sums up over the trajectory we can simply use thresholding).
> - The mechanism needs to be numerically tractable.
>
> Generally, there might be different metrics fulfilling these requirements. However, we consider it nontrivial to come up with a method building on a different metric that fulfills all of the requirements above.

---

> ### Author Response · Authors · 2024-11-27
>
> Thank you for your reply and for considering our revision.
>
> From what we understood, the main concern mentioned in your review was insufficient ablation studies concerning critic overfitting in the naive integration scenario.
>
> We were hoping to address this by adding Appendix E4 to the revised version. Here, we provide several ablations that clearly show when overfitting occurs and that this is a problem of the downstream Q-learning architecture and not the presented rollout scheme.
>
> Do you agree that this part is now clearer and does not leave ambiguity?

---

### Official Review · Reviewer_j8sE · 2024-11-04

**Soundness:** 2
**Presentation:** 2
**Contribution:** 3
**Rating:** 5
**Confidence:** 2

**Summary:**

This paper proposes Infoprop, a rollout mechanism designed to improve sample efficiency and asymptotic performance in model-based reinforcement learning (MBRL). Infoprop addresses the issue of epistemic model uncertainty typically introduced by ensemble methods or dropouts. It projects sampled next states into the maximum likelihood distribution of the predicted next state distribution and uses an information-theoretic criterion to determine the termination point of rollouts. When applied to Dyna-style MBRL, Infoprop demonstrates state-of-the-art performance, as shown in experiment results.

**Strengths:**

1. **Significance**: Model error remains a significant challenge in MBRL, and Infoprop addresses it through the rollout process.
2. **Extensive Experimentation**: The authors provide comprehensive experiments and analysis, including detailed evaluations of algorithm trajectories and environment buffers, which validate the effectiveness of the Infoprop algorithm.

**Weaknesses:**

1. **Assumptions**: The approach relies on two key assumptions. It is important to empirically assess the extent to which these assumptions hold in experimental environments.
2. **Organization**: The theoretical section could benefit from clearer structure. Presenting the main results in the form of definitions or theorems could improve readability. Simplifying technical details and focusing on the implementation—particularly regarding Equation (14)—would make the logical flow smoother.

**Questions:**

1. Is the model error come solely from epistemic noise?
2. What is the advantage of using AES models, and why do you believe Infoprop does not impact their effectiveness?

---

> ### Author Response · Authors · 2024-11-18
>
> **W1:** Thank you for pointing this out. I would kindly refer you to point 2 of our general rebuttal.
>
> **W2:** We agree and have addressed this in the revised version (see points 1 and 3 of the general rebuttal).
>
> **Q1:** Yes, in our set of assumptions that hold well in practice as discussed above, epistemic noise is the only source of model error.
>
> **Q2:** The major benefit of AES models over other model architectures is that they can distinguish aleatoric uncertainty due to process noise from epistemic uncertainty due to model error. This allows to model stochastic systems while having a notion of model uncertainty. However, AES models combined with the standard TS rollout [1] are only capable of making a reliable distinction of this kind for a single transition. As depicted in Figure 1, the TS mechanism mixes both types of uncertainty when using the model for multi-step predictions. Infoprop strengthens the core benefit of AES models (namely the separation of aleatoric and epistemic uncertainty) as it enables this separation for multi-step rollouts instead of just single-step predictions.
>
>     [1] Deep Reinforcement Learning in a Handful of Trials using Probabilistic Dynamics Models, Chua et al. 2018

---

> > ### Author Response · Authors · 2024-11-27
> >
> > Thank you for pointing out concerns about the presentation of the initial version.
> >
> > We would love to hear about what you think about the presentation of the revised version and Section 4 in particular.
> >
> > Do you agree that the presentation is now substantially clearer?

---

> > > ### Comment · Reviewer_j8sE · 2024-11-30
> > >
> > > Thank you for your efforts in revising the paper and providing detailed rebuttals. I have two follow-up questions concerning Figure 1 and the AES model:
> > >
> > > 1. Could you clarify how the data in Figure 1, particularly in the "Trajectory Sampling" plot, was generated? Specifically, how was the model for trajectory sampling trained? It would be more compelling if the data came from a "checkpoint" of the model used during the experiments. Otherwise, reliance on a well-tuned model that eliminates biases might weaken the support for the stated assumptions.
> > >
> > > 2. Could you provide more details on the origin of epistemic error in the AES model?

---

> > > > ### Author Response · Authors · 2024-11-30
> > > >
> > > > Thank you for your follow-up questions. The respective answers are provided below.
> > > >
> > > > 1) The details on how the model in Figure 1 was trained are provided in Appendix B. I will quickly summarize it here: We performed 1000 rollouts with the ground truth dynamics, stored the transition data and trained the PE model for 4 epochs with the parameters provided in Table 1. Using this model we performed 1000 rollouts with either the trajectory sampling rollout scheme (Algorithm 2 in Appendix C) or the Infoprop scheme (Algorithm 1 on page 7).
> > > > Concerning your doubts about the validity of the results: i) Restricting model usage to the space $\mathcal{E}$ via the threshold $\lambda_1$ makes sure the model is only used, where it is sufficiently accurate, such that the assumption is valid. ii) The spread-out distribution of the TS sampling rollouts in Figure 1 shows that the model has not yet completely converged even though it is considered accurate enough. In the limit of infinite data and training i.e. when the model has fully converged, the epistemic uncertainty of the PE model would converge to zero, resulting in identical rollout distributions for TS rollouts and Infoprop rollouts. As we are always limited in data when doing MBRL it is however important to provide good data quality from a non-ideal model, which is the problem Infoprop addresses iii) Figure 4 shows very similar results for a high dimensional problem after a typical training duration. Again, the spread-out data distribution of the TS rollouts indicates that the model has not yet fully converged.
> > > >
> > > > 2) The source of epistemic uncertainty in AES models is the lack of data coverage.
> > > > I build my explanation on the example of a PE model. Each neural network in the ensemble is randomly initialized and trained on a different permutation of the dataset.
> > > > As a consequence, each ensemble member initially predicts a different outcome. In areas of the state action space, where sufficient data was observed, all ensemble members are trained sufficiently well and the ensemble members start to align their predictions. We refer to the disagreement of ensemble members as epistemic uncertainty.
> > > > In the limit of infinite data and training, the epistemic uncertainty will vanish. In model-based RL, however, we typically have very limited data, such that epistemic uncertainty is almost always present.

---

> > > > > ### Comment · Reviewer_j8sE · 2024-12-03
> > > > >
> > > > > Thanks for the rebuttal. I will keep the score.

---

### Author Response · Authors · 2024-11-18
**General Rebuttal**

Dear reviewers, first of all, thank you for the helpful, constructive feedback. We have uploaded a revised version of the paper. Here the first 26 pages correspond to the new paper with appendix. In case you prefer reading the diff file, this corresponds to page 27 onwards.  We hope the new version can address your concerns. In the following, we want to address concerns that were mentioned by several reviewers. Replies to individual requests can be found in the respective individual rebuttals.
We are grateful for the excellent comments, which have substantially improved the readability of our paper.

**1) Section 4 is hard to follow:**
We have substantially revised the presentation of Section 4 according to your comments, i.e. we chose a more formal presentation that provides the core results in definition, lemma, and theorem environments while moving technicalities to the appendix. In particular, the key results of the Infoprop rollout are presented in Theorem 1 and the key idea of Section 4 is given at the beginning supported by a figure that builds intuition.


**2) Meaning/Soundness of Assumptions not discussed.**
The last paragraph of Section 4.1. (end of page 4) now discusses the assumptions and why they are valid empirically. From model rollouts e.g. in Figure 1, we see the models do not underestimate stochasticity due to aleatoric uncertainty and have negligible bias.

**3) Presentation of the Infoprop state (former Eq.14) is confusing.**
We have decided to remove unnecessary technical details from the presentation of the Infoprop state in Definition 5. Instead, we provide an additional step-by-step derivation of the Infoprop state in Appendix D3.
Thus, we aim to convey a better intuition, while still providing the technical details.

---

### Meta-Review · Area_Chair_Y7tG · 2024-12-22

**Metareview:**

This paper proposes Infoprop, an original method with a rollout mechanism designed to improve sample efficiency and asymptotic performance in model-based reinforcement learning.
The reviewers appreciate the theoretical foundation of the method, the originality of the method, as well as the presented experiments.
There are some concerns about having a more thorough evaluation of the limitation of the method, particularly regarding the reported instabilities.
There are some comments regarding the writing style, although these could be improved and are not fundamental about the method.

**Additional Comments On Reviewer Discussion:**

Some of the reviewers concerns have been addressed

---

### Decision · Program_Chairs · 2025-01-22

Accept (Poster)